# Massively Parallel Environments for Large-Scale Combinatorial Optimizations Using Reinforcement Learning

## Abstract

Most combinatorial optimization (CO) problems are NP-hard and difficult to find high-quality solutions. Reinforcement learning (RL) is a promising technique due to its powerful search capability; however, sampling speed is a common bottleneck. Current benchmark works only provide instance-wise approaches, while our work cover both instance-wise and distribution-wise approaches, especially in large-scale CO problems. In this paper, we build 24 GPU-based massively parallel environments for 12 CO problems, i.e., each problem has two environments; and use them to train RL-based approaches. We reproduce benchmark RL algorithms, including instance-wise and distribution-wise approaches especially in large-scale CO problems, on both synthetic datasets and real-world datasets. Take the graph maxcut problem as an example. The sampling speed is improved by at least two orders over conventional implementations, and the scale (i.e., number of nodes) of trained problems in a distribution-wise approach is up to thousands of nodes, i.e., improved by one order. The objective value obtained by inference ($100 \sim 200$ seconds) in the distribution-wise scenario is almost the same as the state-of-the-art (SOTA) solver Gurobi (running for 1 hour), and better than the SOTA RL-based approach. The code is available at: https://github.com/OpenAfterReview.

## 1 Introduction

Combinatorial optimization (CO) problems are critical in many domains, such as financial portfolio allocation Hautsch & Voigt (2019); Zhou et al. (2019); Liu et al. (2022), smart grids Chakraborty et al. (2017); Morstyn (2022), and transportation Zhu et al. (2016; 2019). These problems are generally NP-hard, which is difficult to obtain high-quality solutions due to the exponential growth of the solution space with problem size. Conventional methods such as heuristics Atkinson (1994) and evolutionary algorithms Cai et al. (2014) often fail to find good solutions in a short time frame. Deep reinforcement learning (RL) Mazyavkina et al. (2021); Bengio et al. (2021); Liu et al. (2021) offers a promising approach to solving CO problems for three advantages. First, RL possesses powerful search capabilities especially in large solution space. Second, RL leverages GPUs for training neural networks Makoviychuk et al. (2021). Third, RL can learn automatic heuristics Dai et al. (2017); Chen et al. (2023); Sun et al. (2023) from data without the need to hand-craft heuristics on a case-by-case basis.

Sampling is a bottleneck of RL algorithm for solving large-scale CO problems Makoviychuk et al. (2021); Sun et al. (2023). Training the policy network is essentially estimating the gradients via a Markov chain Monte Carlo (MCMC) simulation, which requires a large number of samples from environments. Moreover, the solution space in CO problems increases exponentially with the problem size, which makes the sampling more important. Existing CPU-based environments have two significant disadvantages: 1) The number of CPU cores is typically small, generally ranging from 16 to 256, resulting in a small number of parallel environments. 2) The communication link between CPUs and GPUs has limited bandwidth. The samples obtained on CPUs are sent to GPUs for training a policy network, then the new policy will be returned to CPUs, which is inefficient.

Researchers propose to use GPU-based parallel environments for RL methods Gulino et al. (2024); Lechner et al. (2024); Rutherford et al. (2024); Bonnet et al. (2024); Berto et al. (2023). Our work has

several differences with current works. First, the current works such as Jumanji Bonnet et al. (2024) and RL4CO Berto et al. (2023) only provide instance-wise approaches, ignoring the distribution-wise approaches. Our work cover both instance-wise and distribution-wise approaches. Second, the current works do not cover all typical patterns in this community: Jumanji and RL4CO only provide the methods only in one pattern, which as based on Markov decision process (MDP) model and conventional RL methods such as PPO Schulman et al. (2017) and DQN Dai et al. (2017), ignoring the RL-based annealing methods. The experiments in Section 5 show that the performance of annealing methods is generally better.

To address the bottleneck of sampling speed, we use GPU-based massively parallel environments that have several advantages. 1) There are several thousands of cores on a GPU that can be used to build massively parallel environments. Our implementations allow up to $32,768$ parallel environments. 2) The communication between CPUs and GPUs is bypassed, since the samples are stored as GPU tensors and thus readily available for training policy networks. 3) The RL-based methods using GPU-based parallel environments in two important patterns of this community are implemented, which cover most of existing RL methods. Users can implement their RL methods using parallel environments easily following the common instructions. 4) With the help of GPU-based parallel environments, RL-based distribution-wise approaches can be trained on large-scale CO problems.

Our contributions are summarized as follows:

- We build $24$ GPU-based massively parallel environments for $12$ CO problems, i.e., each problem has two environments, inspired by Isaac gym Makoviychuk et al. (2021) (i.e., physical simulations for robot learning). Our implementations allow up to $32,768$ parallel environments. The communication bottleneck between CPUs and GPUs is bypassed.

- We reproduce benchmark RL algorithms using GPU-based massively parallel environments, including both instance-wise and distribution-wise approaches, on both synthetic datasets and real-world datasets. These approaches cover most important patterns in this community. We will add new RL algorithms if they spring in the future which fall in these patterns. The results demonstrate a significant speedup in both training efficiency and sampling speed.

- By utilizing GPU-based massively parallel environments, RL-based distribution-wise approaches can be trained on large-scale CO problems, e.g., thousands of nodes compared with hundreds of nodes in current works. That is, the scale is improved by one order.

- In a distribution-wise approach supported by massively parallel environments, the objective value by inference ($100 \sim 200$ seconds) is almost the same as Gurobi (running for 1 hour) and better than the current SOTA distribution-wise approach.

The remainder of this paper is organized as follows. Section 2 describes two formulations of CO problems. Section 3 describes existing methods for CO problems. In Section 4, we develop massively parallel environments for RL algorithms. Section 5 demonstrates the performance.

## 2 COMBINATORIAL OPTIMIZATION PROBLEMS

Combinatorial optimization (CO) problems aim to find a high-quality solution from a large search space, where the number of feasible solutions grows exponentially with the problem size. There are two common formulations for CO problems: integer linear programming (ILP) model and quadratic unconstrained binary programming (QUBO) model (a.k.a., Ising model formulation).

### 2.1 ILP FORMULATION

Integer linear programming (ILP) is a standard formulation Ibaraki (1976) with the *canonical form*:

$$
\begin{aligned}
\min \; & \boldsymbol{c}^T \boldsymbol{x} \\
\text{s.t. } & \boldsymbol{A}\boldsymbol{x} \leq \boldsymbol{b}, \quad \boldsymbol{x} \geq 0, \quad \boldsymbol{x} \in \mathbb{Z}^n,
\end{aligned}
\tag{1}
$$

where $\boldsymbol{x}$ is a vector of $n$ decision variables, $\boldsymbol{c}$ is a vector of $n$ coefficients, $\boldsymbol{A} \in \mathbb{R}^{m \times n}$ and $\boldsymbol{b} \in \mathbb{R}^m$ together denote $m$ linear constraints, and $\boldsymbol{x} \in \mathbb{Z}^n$ implies that we are interested in integer solutions.

## 2.2 QUBO FORMULATION

Consider a 1D Ising model Cipra (1987) with a ring structure and an external magnetic field $h_i$, there are $N$ nodes with $(N + 1) = 1 \mod N$; a node $i$ has a spin $\boldsymbol{x}_i \in \{+1, -1\}$ (where $+1$ for up and $-1$ for down). Two adjacent sites $i$ and $i + 1$ have an energy $w(i, i + 1)$ or $-w(i, i + 1)$ if they have the same direction or different directions, respectively.

The whole system will evolve into the ground state with the minimum Hamiltonian:

$$\min \; f(\boldsymbol{x}) = -\sum_{i=1}^{N} h_i \boldsymbol{x}_i \; - \alpha \; \sum_{i=1}^{N} w(i, i+1) \boldsymbol{x}_i \boldsymbol{x}_{i+1}, \tag{2}$$

where $\alpha > 0$ is a weight, the first term is defined on nodes, and the second term is defined on nodes' interactions.

## 3 EXISTING METHODS FOR CO PROBLEMS

### 3.1 CLASSICAL METHODS

To solve CO problems, researchers may first formulate them as integer linear programming (ILP). Then ILP is relaxed to linear programming (LP), i.e., change integer variables to continuous ones. And then LP solutions are obtained by using the simplex method Nash (2000). However, the LP solutions may not be all integers; and therefore, the following two methods are used to obtain integer solutions.

**Branch-and-bound** method Brusco et al. (2005) iteratively selects a non-integer variable to branch, i.e., dividing the original problem into two smaller subproblems, and then chooses a branch that may result in a good solution. This process iterates until a good integer solution is obtained.

**Cutting plane** method Marchand et al. (2002) adds a series of cutting planes (linear constraints) so that the feasible region of the relaxation problem is reduced, but all the feasible integer solutions are retained. Finally, the original constraints and the added cutting planes constitute a minimum convex hull so that we obtain the optimal integer solution by directly solving the new LP using the simplex method Nash (2000).

### 3.2 HEURISTIC METHODS

**Greedy algorithm** starts from empty set $S = \emptyset$ and will construct a solution by sequentially adding nodes to a partial solution $S$, based on maximizing some evaluation function $Q(\cdot)$ that measures the quality of a node in the context of the current partial solution.

**Local search** starts with an initial solution and searches a better neighborhood solution. It improves the current solution iteratively: it replaces the current solution and the search continues if such a solution is found, and returns a locally optimal solution otherwise.

**Simulated annealing** Dowsland & Thompson (2012) is a metaheuristic to approximate global optimization. It is based on the analog of metal's cool and anneal. It selects a better/worse neighboring state with a high/low acceptance probability over many neighboring states according to the current temperature. This prevents being stuck in local optima. The temperature decreases and finally converges over iterations, i.e., reaching a solid state with minimum energy.

### 3.3 REINFORCEMENT LEARNING METHODS

**RL-driven heuristics** such as greedy Dai et al. (2017), k-opt Ma et al. (2024), beam search Choo et al. (2022) and local search Chen et al. (2023) can learn heuristics from data automatically. Dai et al. (2017) propose an approach to learn a greedy policy that exploit the structure of such recurring problems, which combines RL and graph embedding. They build MDP models for CO problems, and then combine graph embedding Dai et al. (2016) and RL to obtain the policy. The solution starts from a partial solution, and a new node is added iteratively until a whole solution is obtained. In the traveling salesman problem (TSP), Fu et al. (2021) propose to train a small-scale model, and then build heat maps for instances with larger size. Chen et al. (2023) propose a Monte carlo

policy gradient method for binary optimization, which combines RL with local search to improve the quality when searching better neighborhood nodes. Choo et al. (2022) propose a simulation-guided beam search method, which examines solutions within a fixed-width tree search that both a neural network-learned policy and a simulation identify as promising. Ma et al. (2024) propose a learning to search the k-opt method which focuses on the routing problem, so that it can perform flexible k-opt exchanges. Most of the above works target on how to use RL-based methods for CO problems, ignoring the effectiveness of massively parallel environments. Therefore, the size of trained problems may be small, e.g., hundreds of nodes in graph maxcut problem or routing problems.

Current RL-based methods for CO problems are classified to two categories: instance-wise and distribution-wise scenarios. In the instance-wise scenario, also named as end-to-end, RL-based methods solve the problem case by case. In the distribution-wise scenario, RL-based methods train neural networks based on samples, and then obtain the solution by inference. Both scenarios require a large number of samples, especially the distribution-wise scenario since it requires the samples over all the distribution. The scale of trained problems in current research works is generally small, e.g., 20~50 nodes in TSP Fu et al. (2021) or 50~100 nodes in the graph maxcut problem Dai et al. (2017). With the support of GPU-based massively parallel environments, the performance of current RL-based methods, especially the trained problem size in the distribution-wise scenario, can be significantly improved.

**GPU-based massively parallel environments for RL methods** can significantly improve the sampling speed for training. Gigastep Lechner et al. (2024) and JaxMARL Rutherford et al. (2024) are two typical multi-agent RL (MARL) libraries with parallel environments. The RL4CO library Berto et al. (2023) provides a unified framework for RL-based CO algorithms. Jumanji Bonnet et al. (2024) presents a diverse suite of scalable reinforcement learning environments in JAX. All the above research works show the significance of GPU-based parallel environments in RL. Jumanji and RL4CO provide only instance-wise methods for CO problems, and Gigastep and JaxMARL only provide MARL algorithms without using them for CO problems. Pgx Koyamada et al. (2023) and Craftax Matthews et al. (2024) focus on parallel simulators for games. Moreover, RL-based annealing algorithms such as Chen et al. (2023); Liu & Zhang (2024); Lu & Liu (2023); Li et al. (2021); Lu et al. (2023) are typical methods, which fall in an important pattern that we propose in Section 4, while the above works do not cover them. Different from the above benchmark works, we use GPU-based parallel environments in RL for CO problems over all important patterns, including instance-wise and distribution-wise approaches, especially in large-scale problems.

## 4 MASSIVELY PARALLEL CO ENVIRONMENTS ON GPUS

Table 1: Existing RL algorithms follow two patterns.

|  | Pattern I | Pattern II |
|---|---|---|
| Space | MDP: $(\mathcal{S}, \mathcal{A}, \mathcal{R}, \mathcal{P}, \gamma)$ | $(\mathcal{S}, f)$ |
| Deterministic | No | Yes |
| Initial distribution | $d(s_0)$ | $d(s_0)$ |
| Reward | $r(s, a) = f(s \bigcup \{a\}) - f(s)$ | $r(s, s') = f(s')$ |
| Policy | $\pi(a\|s) : \mathcal{S} \longrightarrow \mathcal{A}$ | $\pi(s'\|s) : \mathcal{S} \longrightarrow \mathcal{S}$ |
| Transition | $\mathcal{S} \times \mathcal{A} \longrightarrow \mathcal{S}$ | $\mathcal{S} \longrightarrow \mathcal{S}$ |
| Trajectory | $\tau = (s_0, a_0, r_1, \ldots, s_T, a_T, r_{T+1})$ | $\tau = (s_0, s_1, \ldots, s_T)$ |
| Trajectory probability | $P(\tau) = d(s_0)\Pi_{t=1}^{T}[\pi(a_{t-1}\|s_{t-1}) \times P(s_t\|s_{t-1}, a_{t-1})]$ | $P(\tau) = d(s_0)\Pi_{t=1}^{T}\pi(s_t\|s_{t-1})$ |

For RL algorithms, the sampling is based on Monte Carlo Markov chain (MCMC) simulations. We build massively parallel environments on GPUs. In particular, we propose gym-style templates of building parallel environments: all components such as parallel solutions and objective values are stored as GPU tensors, thus all operations are executed parallelly on GPUs.

**Example**. Take the graph maxcut problem as an example, we assume the graph has $n$ nodes, and we build $k$ environments on one GPU. To enable parallel operations, we use a tensor of size $k \times n$ to store all the solutions for these parallel environments. All the operations are executed parallelly using PyTorch Paszke et al. (2019), and if we use $m$ GPUs, it will support $k \times m$ environments.

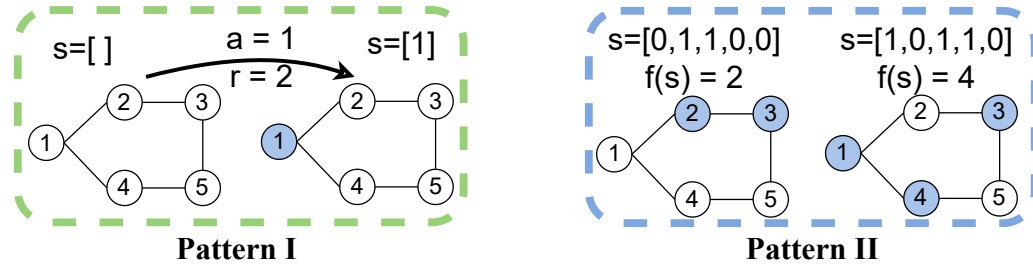

Figure 1: Two patterns for graph maxcut.

We introduce two typical patterns for massively parallel MCMC simulations:

- **Pattern I**: RL-based heuristic Dai et al. (2017); Sun et al. (2023) formulates the CO problem as Markov decision process (MDP), and then use RL algorithms to select the node with the maximum Q-value and add it into the solution set. Take graph maxcut as an example. In left part of Fig. 1, the initial solution set (a state) is empty, i.e., no node is selected, and then we select the node 1 with the maximum Q-value and add it into the solution set.

- **Pattern II**: policy-based methods Liu & Zhu (2023); Mohseni et al. (2022); Liu & Zhang (2024); Lu & Liu (2023) first formulate the CO problem as a QUBO problem, and then learn a policy using say REINFORCE algorithm Zhang et al. (2021) to minimize the Hamiltonian Liu & Zhu (2023); Liu & Zhang (2024) objective function. Take graph maxcut as an example. In right part of Fig. 1, the initial solution set has node 2 and 3, and then it transition to a new solution set with node 1 and 4.

Table 1 compared the above two patterns in detail. $\gamma$ is the discount factor, and $\theta$ is the parameters of neural networks, and $f$ is the objective function of CO problems.

The two patterns have several differences. First, their application scopes are different. The application scope of Pattern I may be smaller than Pattern II since it may not handle the problems which can not be modeled by adding node one by one, e.g., the portfolio optimization problem. Pattern II has wider application scope since most CO problems can be formulated by QUBO. Second, the functionalities of the neural works are different. In Pattern I, the neural works are used for approximating the policy and Q-value. While in Pattern II, the neural works are only used for approximating the policy. Third, the sampling speed in Pattern I is generally faster than Pattern II since there are some complex functions in Pattern II. Moreover, from experiments, we see that the methods in Pattern II are generally better than that in Pattern I.

### 4.1 PATTERN I FOR CO ENVIRONMENTS

The objective function and gradient are

$$J_I(\theta) = \mathbb{E}\left[\sum_{t=0}^{T} r(s_t, a_t) + \gamma Q'(s_{t+1}, \pi(s_{t+1}))\right], \quad (3)$$

$$\nabla_\theta J_I(\theta) = \nabla_\theta \mathbb{E}\left[\sum_{t=0}^{T} r(s_t, a_t) + \gamma Q'(s_{t+1}, \pi(s_{t+1}))\right], \quad (4)$$

where $s$ is the state, $r$ is the reward function, $\pi$ is the policy, and the $T$ is the length of the trajectory.

There are three important functions for a gym-style environment:

- reset(): Set the selected nodes as an empty set.
- step(): Select the node with maximum Q-value and then add it to the set.
- reward(): Calculate the objective values over all simulation environments.

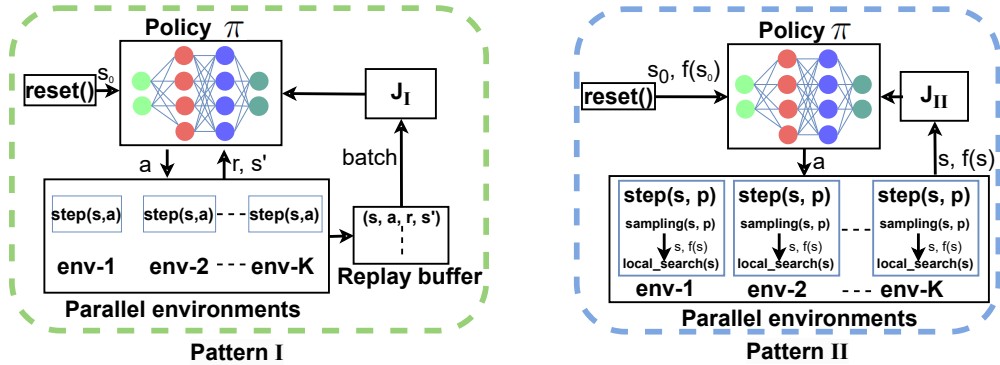

Figure 2: Massively parallel environments for two patterns.

Take graph maxcut as an example, we present the Markov decision process (MDP).

- State $\mathcal{S}$: a set of selected nodes on a graph.
- Transition $\mathcal{S} \xrightarrow{a} \mathcal{S}'$: is deterministic, and the new state $\mathcal{S}'$ will be $\mathcal{S} \bigcup \{a\}$ if the action $a$ is taken based on the current state $\mathcal{S}$.
- Action $\mathcal{A}$: is to select a node in another set on a graph.
- Reward $R(\mathcal{S}, a)$: is the the new cut value after taking action $a$ (adding a node to the state $\mathcal{S}$) minus the current cut value.
- Policy $\pi(a|\mathcal{S})$: is to select a node $a$ to add to the the set or state $\mathcal{S}$, which obtains a reward $R(\mathcal{S}, a)$.

### 4.2 PATTERN II FOR CO ENVIRONMENTS

The objective function and gradient are

$$J_{II}(\theta) = \mathbb{E}\left[\sum_{t=1}^{T} f(s_t)\right], \quad \nabla_\theta J_{II}(\theta) = \nabla_\theta \mathbb{E}\left[\sum_{t=1}^{T} f(s_t)\right], \tag{5}$$

where $f$ is the objective function and the $T$ is the length of the trajectory.

We introduce four important functions Dai et al. (2017); Sun et al. (2023) for a gym-style environment:

- reset(): Generate a random initial solution.
- step(): Search for better solutions based on the current solution. It has two sub-functions.
    - sampling() is the sampling method.
    - local_search() returns a better solution by flipping some bits. It can improve the quality of the current solution in a local domain.
- pick_good_xs(): Select the good solutions in all parallel environments, where each environment returns exactly one good solution with corresponding objective value.
- obj(): Calculate the objective value.

Take graph maxcut as an example, we list the training process.

- Initialize a random solution $\mathcal{S}$.
- Select some nodes and flip them, and obtain a new solution $\mathcal{S}'$, and then calculate the cut value $f'$. Execute the above operation for plenty of times. Through the sampling method to filter the samples. Use the local search method to improve the quality of samples, e.g., the cut value improves from $f'$ to $f''$.

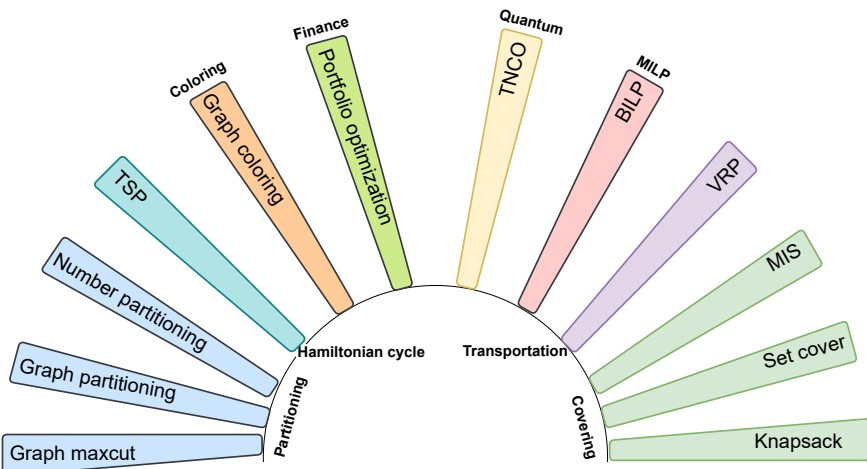

Figure 3: Typical CO problems.

- In each environment, reserve the sample with the maximum cut value.
- Update the gradient based on the reserved samples.

With respect to the objective to optimize for neural networks, we set it as the cut value plus a weighted entropy, and the temperature decreases with iterations. Finally, the algorithm converges and we obtain the maximum cut value.

## 5 PERFORMANCE EVALUATION

We select 12 typical CO problems over 8 areas in Fig 3. The ILP and QUBO formulations of these problems are moved to the Appendix for limited pages. We implement the algorithms on a DGX-2 server with NVIDIA A100 GPUs. Take graph maxcut as an example, we show the performance. We use several datasets: Gset [1], and generated graphs in three distributions: barabasi albert (BA), erdos renyi (ER), and powerlaw (PL).

### 5.1 IMPROVING THE QUALITY OF SOLUTIONS AND CONVERGENCE

GPU-based parallel environments can significantly improve the quality of solutions during training, since RL methods require many high-quality samples from the environments. Take graph maxcut as an example. We select G22 in the Gset dataset. The number of epochs is set to 30, and the number of steps in each epoch is set to 24. Fig. 4 shows the objective values vs. number of epochs with different number of GPU-based parallel environments in MCPG Chen et al. (2023), which belongs to Pattern II. We see that, when the number of environments is 1, the highest objective value is 13,257, and the convergence is very slow; however, when the number of environments is $1,024 \sim 4,096$, the highest objective value is approaching the best-known solution 13,359, and the convergence is very fast. Therefore, generally, the more parallel environments, the higher objective values, and the faster convergence.

### 5.2 IMPROVING THE SPEED OF DATA SAMPLING

We use 256 environments on CPU, and 512, 1024, 2048, 4096, 8192, 16384 and 32768 environments on GPU to test the speedup, and the length of the trajectory is set at 512, 256, 128, 64, 31, 16, and 8 correspondingly. We test graph maxcut on a BA graph with 1000 nodes and 3984 edges. We select S2V-DQN Dai et al. (2017) in Pattern I, and MCPG Chen et al. (2023) in Pattern II. From Fig. 5, we see that the reward or objective value with more environments changes much faster than others.

---

[1]Gset: `https://web.stanford.edu/~yyye/yyye/Gset/`

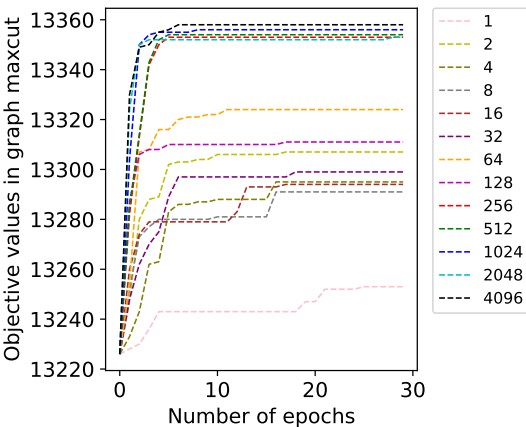

Figure 4: Objective values in graph maxcut vs. number of epochs with different number of GPU-based parallel environments in Pattern II.

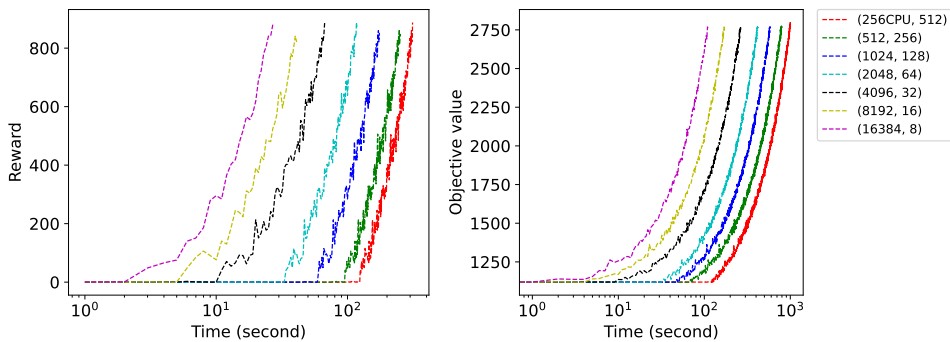

Figure 5: Reward/objective vs. time for graph maxcut (left: Pattern I, right: Pattern II)

Fig. 6 shows the speedup for graph cut in two patterns. The speedup with the most environments is two orders compared with that with the least environments. The number of collected samples per second in Pattern I is larger than Pattern II since there exist several complicated processes such as the sampling algorithms in Pattern II.

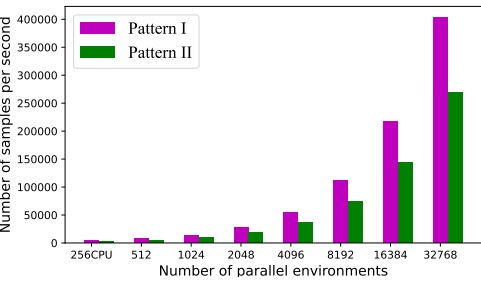

Figure 6: Speed of data sampling in graph maxcut

## 5.3 RESULTS FOR INSTANCE-WISE SCENARIO

In Table 2, we selected 10 instances in each dataset, and we implemented 11 algorithms: Greedy (Approximation ratio = 2), SDP (Approximation ratio = $\frac{1}{0.878}$), simulated annealing (SA), genetic algorithm (GA), Gurobi (QUBO) [2], S2V-DQN Dai et al. (2017) in Pattern I, physical-inspired GNN (PI-GNN) Schuetz et al. (2022) in Pattern I, iSCO Sun et al. (2023) in Pattern II, dREINFORCE Liu & Zhang (2024); Lu & Liu (2023) in Pattern II, MCPG Chen et al. (2023) in Pattern II, and Jumanji Bonnet et al. (2024) in Pattern I. Considering that Jumanji has not implemented the RL method for the graph maxcut problem, we implemented it using the PPO algorithm Schulman et al. (2017) with GPU-based parallel environments, following the codes of TSP in Jumanji. We see that the methods in Pattern II are generally better than Pattern I, since some techniques (e.g., sampling algorithms) are added to improve the quality. With respect to the Gurobi method (running for 1 hour), and it is more easily to obtain better results using QUBO than MILP; therefore, we only present the results using QUBO. The best objective value is denoted by boldface. The results on ER and PL distributions are moved to the Appendix.

Table 2: Results for graph maxcut on BA distribution in instance-wise scenario

| Nodes | Greedy | SDP | SA | GA | Gurobi | S2V-DQN (Pattern I) | PI-GNN (Pattern I) | iSCO (Pattern II) | dREINFORCE (Pattern II) | MCPG (Pattern II) | Jumanji (Pattern I) |
|---|---|---|---|---|---|---|---|---|---|---|---|
| 100 | 272.1 | 272.5 | 272.6 | **284.1** | **284.1** | 283.1 | 273.0 | **284.1** | **284.1** | **284.1** | **284.1** |
| 200 | 546.9 | 552.9 | 552.0 | 582.9 | **583.0** | 580.8 | 560.6 | 581.5 | **583.0** | **583.0** | **583.0** |
| 300 | 833.2 | 839.3 | 834.7 | **880.4** | **880.4** | 875.1 | 846.3 | 877.2 | **880.4** | **880.4** | **880.4** |
| 400 | 1112.1 | 1123.9 | 1116.4 | 1180.9 | 1180.4 | 1175.3 | 1174.6 | 1176.5 | **1181.9** | 1179.5 | **1181.9** |
| 500 | 1383.8 | 1406.3 | 1387.7 | 1477.7 | 1476.0 | 1453.4 | 1436.8 | 1471.3 | **1478.3** | **1478.3** | 1471.3 |
| 600 | 1666.7 | 1701.2 | 1670.6 | 1780.3 | 1777.0 | 1730.7 | 1768.5 | 1771.0 | **1781.5** | 1778.6 | 1779.2 |
| 700 | 1961.9 | 1976.7 | 1966.0 | 1989.2 | 2071.2 | 2038.9 | 1989.4 | 2070.2 | **2076.6** | **2076.6** | 2071.3 |
| 800 | 2237.9 | 2268.8 | 2244.9 | 2375.5 | 2358.9 | 2333.7 | 2365.9 | 2366.9 | **2377.8** | 2372.9 | 2373.4 |
| 900 | 2518.1 | 2550.3 | 2524.8 | 2670.1 | 2658.3 | 2614.9 | 2539.7 | 2662.4 | **2675.1** | 2670.6 | 2671.2 |
| 1000 | 2793.8 | 2834.3 | 2800.8 | 2967.9 | 2950.2 | 2906.3 | 2846.8 | 2954.0 | **2972.3** | 2968.7 | 2951.4 |

Table 3: Results for graph maxcut on the Gset dataset in instance-wise scenario.

| Graph | Nodes | Edges | BLS | DSDP | KHLWG | RUN-CSP | PI-GNN (Pattern I) | iSCO (Pattern II) | dREINFORCE (Pattern II) | MCPG (Pattern II) | Jumanji (Pattern I) |
|---|---|---|---|---|---|---|---|---|---|---|---|
| G14 | 800 | 4694 | **3064** | - | 2922 | 3061 | 2943 | 3056 | **3064** | **3064** | **3064** |
| G15 | 800 | 4661 | **3050** | 2938 | **3050** | 2928 | 2990 | 3046 | **3050** | **3050** | 2979 |
| G22 | 2000 | 19990 | **13359** | 12960 | **13359** | 13028 | 13181 | 13289 | **13359** | **13359** | 13261 |
| G49 | 3000 | 6000 | **6000** | **6000** | **6000** | **6000** | 5918 | 5940 | **6000** | **6000** | 5987 |
| G50 | 3000 | 6000 | **5880** | **5880** | **5880** | **5880** | 5820 | 5874 | **5880** | **5880** | 5872 |
| G55 | 5000 | 12468 | 10294 | 9960 | 10236 | 10116 | 10138 | 10218 | **10298** | 10296 | 10283 |
| G70 | 10000 | 9999 | 9541 | 9456 | 9458 | - | 9421 | 9442 | **9586** | 9578 | 9554 |

Table 3 presents the results of 9 approaches in seven instances from Gset: breakout local search (BLS) Benlic & Hao (2013), SDP (DSDP) Choi & Ye (2000), Tabu search (KHLWG) Kochenberger et al. (2013), recurrent GNN (RUN-CSP) Toenshoff et al. (2021), PI-GNN Schuetz et al. (2022), iSCO Sun et al. (2023), dREINFORCE Liu & Zhang (2024); Lu & Liu (2023), MCPG Chen et al. (2023), and Jumanji Bonnet et al. (2024). We see that RL methods (e.g., dREINFORCE and MCPG) of Pattern II demonstrate clearer advantages over RL methods of Pattern I (e.g., S2V-DQN and Jumanji) and conventional methods (e.g., DSDP), especially in larger instances.

## 5.4 RESULTS FOR DISTRIBUTION-WISE SCENARIO

Considering of limited scope, we only show the results of the graph maxcut problem, which sufficiently reveal the effectiveness of massively parallel environments. In each dataset, we selected 30 instances. To clearly show the comparisons, we introduce a metric: Alg1#Alg2 = $\frac{\text{obj1} - \text{obj2}}{\text{obj2}}$, which is the increasing ratio of the objective value of Alg1 compared with Alg2.

From Tab. 4 and Tab. 5, we see that, with the support of massively parallel environments, the scale of trained graphs and the quality of solutions are improved. For the graphs with nodes ranging from 100 to 1200 nodes, the objective value obtained by inference (100 $\sim$ 200 seconds) is almost the same as Gurobi (running for 1 hour), and improved by 0.05% $\sim$ 5% compared with SOTA distribution-wise approach S2V-DQN Dai et al. (2017). For the graphs with nodes ranging from 2000 to 4000 nodes, the objective value obtained by inference is improved by about 1% compared with Gurobi. The

---

[2]Gurobi: `https://www.gurobi.com`

Table 4: Results for graph maxcut on BA distribution in distribution-wise scenario

| #Nodes | dREINFORCE | Gurobi | dREINFORCE#Gurobi | S2V-DQN#Gurobi | dREINFORCE#S2V-DQN |
|---|---|---|---|---|---|
| 100 | 283.7 | 283.7 | 0 | -1.63% (100 $\sim$ 200) | 1.63% |
| 200 | 583.27 | 583.27 | 0 | -1.79% (200 $\sim$ 300) | 1.79% |
| 300 | 880.43 | 880.43 | 0 | -1.63% (300 $\sim$ 400) | 1.63% |
| 400 | 1179.70 | 1179.17 | 0.0452% | -1.03% (400 $\sim$ 500) | 1.08% |
| 500 | 1479.53 | 1477.60 | 0.131% | -1.63% (500 $\sim$ 600) | 1.76% |
| 1000 | 2970.50 | 2952.20 | 0.487% | -2.38% (1000 $\sim$ 1200) | 2.87% |
| 1100 | 3265.73 | 3250.17 | 0.264% | -2.38% (1000 $\sim$ 1200) | 2.64% |
| 1200 | 3557.93 | 3547.07 | 0.392% | -2.38% (1000 $\sim$ 1200) | 2.77% |
| 2000 | 4060.92 | 4025.89 | 0.87% | - | - |
| 3000 | 5676.26 | 5624.51 | 0.92% | - | - |
| 4000 | 11942.50 | 11821.92 | 1.02% | - | - |

Table 5: Results for graph maxcut on ER distribution in distribution-wise scenario

| #Nodes | dREINFORCE | Gurobi | dREINFORCE#Gurobi | S2V-DQN#Gurobi | dREINFORCE#S2V-DQN |
|---|---|---|---|---|---|
| 100 | 507.83 | 507.83 | 0 | -0.05% (100-200) | 0.05% |
| 200 | 1858.93 | 1856.13 | 0.151% | -1.05% (200 $\sim$ 300) | 1.20% |
| 300 | 4063.20 | 4062.93 | 0 | -2.65% (300 $\sim$ 400) | 2.66% |
| 400 | 7042.10 | 7041.67 | 0 | -3.59% (400 $\sim$ 500) | 3.60% |
| 500 | 10862.30 | 10862.40 | 0 | -5.77% (500 $\sim$ 600) | 5.77% |
| 1000 | 41735.16 | 41765.87 | -0.0735% | -5.07% (1000 $\sim$ 1200) | 5.00% |
| 1100 | 50219.47 | 50286.8 | -0.134% | -5.07% (1000 $\sim$ 1200) | 4.94% |
| 1200 | 59506.30 | 59561.83 | -0.093% | -5.07% (1000 $\sim$ 1200) | 4.98% |
| 2000 | 163587.11 | 162111.89 | 0.91% | - | - |
| 3000 | 363365.19 | 359767.52 | 1.00% | - | - |
| 4000 | 643101.65 | 634848.62 | 1.30% | - | - |

scale of trained graphs for distribution-wise approaches is 2,000~4,000 nodes, and is improved by one order compared with 50~100 nodes of current research works Dai et al. (2017); Drakulic et al. (2024).

The reasons why massively parallel environments supported approach (dREINFORCE) has better performance (in terms of size of graphs and quality of solutions) than S2V-DQN and Gurobi are as follows. First, by using massively parallel environments, we can obtain many many samples for training. Second, the obtained samples are of high-quality since RL has powerful search skills with the help of GPUs, so better policy can be obtained. Moreover, we use several tricks to improve the quality of solutions, e.g., sampling algorithms. Gurobi may obtain worse solutions when the size of problems increases, since its knowledge/heuristic may not work well in larger instances. Third, we use graph auto-encoder Kipf & Welling (2016); Fan et al. (2021) to learn the topology and meaningful representations of graphs, and use decoder to predict the probability distribution.

## 6 CONCLUSION

We noticed that sampling is a bottleneck for large-scale combinatorial optimization (CO) problems using reinforcement learning (RL) algorithms; therefore, we propose to use GPU-based massively parallel environments to speed up the sampling process. We build a benchmark, including instance-wise and distribution-wise approaches, using GPU-based massively parallel environments (say up to 32,768 environments). The results demonstrate that the sampling speed is increased by at least two orders. The scale (i.e., number of nodes) of trained problems in distribution-wise approaches increases one order, and the performance is almost the same as Gurobi (running for 1 hour) and better than the SOTA distribution-wise approach.

However, if the CO environments are very complicated, the operations on tensors may be hard and they require large memory for GPUs. We only implemented parallel environments on typical CO problems, and the implementation on real-world complicated CO problems will be done in the future, including the large-scale dynamic ridesharing problem in transportation, power scheduling in smart grids, portfolio allocation in finance, and supply chain optimization in industrial internet.

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
