# A   DATASETS

## A.1   GRAPH DATA

Graph data includes Gset, synthetic datasets in three distributions: barabasi albert (BA), erdos renyi (ER), and powerlaw (PL), and TSPLIB instances.

## A.2   NON-GRAPH DATA

Non-graph data includes the knapsack dataset, set cover dataset, etc.

# B   MORE RESULTS FOR GRAPH MAXCUT PROBLEM IN INSTANCE-WISE SCENARIO

Table 6: Results for graph maxcut on erdos renyi (ER) distribution

| Nodes | Greedy | SDP | SA | GA | Gurobi | S2V-DQN (Pattern I) | PI-GNN (Pattern I) | iSCO (Pattern I) | dREINFORCE (Pattern II) | MCPG (Pattern II) | Jumanji (Pattern I) |
|---|---|---|---|---|---|---|---|---|---|---|---|
| 100 | 489.3 | 486.5 | 490.2 | **507.1** | **507.1** | 506.3 | 491.6 | **507.1** | **507.1** | **507.1** | **507.1** |
| 200 | 1828.7 | 1810.4 | 1829.5 | **1868.6** | 1866.0 | 1853.4 | 1838.5 | 1867.0 | **1868.6** | 1868.0 | **1868.6** |
| 300 | 3984.2 | 3950.9 | 3986.1 | 4064.3 | 4064.0 | 3986.4 | 3949.3 | 4059.4 | **4069.4** | **4069.4** | **4069.4** |
| 400 | 6905.5 | 6879.6 | 6907.8 | 7035.3 | 7035.0 | 6756.7 | 7024.7 | 7013.1 | **7038.6** | **7038.6** | 6984.9 |
| 500 | 10692.4 | 10609.1 | 10693.9 | 10860.0 | 10858.3 | 10608.4 | 10755.9 | 10830.3 | **10864.7** | 10862.4 | 10861.0 |
| 600 | 15241.0 | 15140.8 | 15242.1 | 15480.4 | 15479.4 | 1514.3 | 15101.4 | 15427.4 | **15482.4** | 15472.5 | 15461.8 |
| 700 | 20627.7 | 20477.0 | 20629.0 | 20876.7 | 20877.6 | 20293.5 | 20603.0 | 20814.9 | **20879.2** | 20868.8 | 20861.2 |
| 800 | 26704.3 | 26492.4 | 26708.2 | 27019.6 | 27020.4 | 26216.3 | 26613.6 | 26932.6 | **27058.9** | 27049.6 | 27045.6 |
| 900 | 33628.4 | 33446.9 | 33630.8 | 34022.7 | 34021.8 | 33051.3 | 33518.7 | 33917.9 | **34073.6** | 34070.4 | 34061.7 |
| 1000 | 41279.6 | 41017.9 | 41281.1 | 41734.1 | 41737.8 | 40753.2 | 41540.3 | 41613.2 | **41796.7** | 41787.1 | 41762.4 |

Table 7: Results for graph maxcut on powerlaw (PL) distribution

| Nodes | Greedy | SDP | SA | GA | Gurobi | S2V-DQN (Pattern I) | PI-GNN (Pattern I) | iSCO (Pattern I) | dREINFORCE (Pattern II) | MCPG (Pattern II) | Jumanji (Pattern I) |
|---|---|---|---|---|---|---|---|---|---|---|---|
| 100 | 268.2 | 270.6 | 268.4 | **282.9** | **282.9** | 278.5 | 271.3 | 282.8 | **282.9** | **282.9** | **282.9** |
| 200 | 548.0 | 554.4 | 550.5 | 578.0 | **578.7** | 563.1 | 572.5 | 578.3 | **578.7** | **578.7** | **578.7** |
| 300 | 824.6 | 833.9 | 826.4 | 877.2 | 877.2 | 859.4 | 843.2 | 876.1 | **879.5** | 878.2 | **879.5** |
| 400 | 1107.1 | 1117.8 | 1110.1 | 1173.2 | 1173.1 | 1152.7 | 1135.8 | 1169.6 | **1178.4** | 1173.9 | **1178.4** |
| 500 | 1386.6 | 1399.0 | 1391.5 | 1471.5 | 1468.1 | 1440.2 | 1455.3 | 1467.0 | **1475.6** | **1475.6** | 1460.4 |
| 600 | 1660.5 | 1683.4 | 1664.1 | 1768.3 | 1760.8 | 1725.6 | 1693.7 | 1758.5 | **1773.9** | 1770.1 | 1769.5 |
| 700 | 1950.8 | 1970.0 | 1955.0 | 2064.9 | 2056.7 | 2004.9 | 1982.4 | 2055.5 | **2067.2** | **2067.2** | 2058.6 |
| 800 | 2228.0 | 2260.7 | 2232.2 | 2361.4 | 2349.8 | 2302.4 | 2346.7 | 2353.4 | **2378.9** | 2375.4 | 2352.7 |
| 900 | 2507.0 | 2540.0 | 2514.1 | 2658.1 | 2643.9 | 2526.4 | 2594.3 | 2645.0 | **2676.1** | 2671.9 | 2667.4 |
| 1000 | 2784.3 | 2817.4 | 2792.7 | 2955.5 | 2942.1 | 2723.3 | 2903.4 | 2949.1 | **2982.8** | 2978.4 | 2963.9 |

From Table 6 and 7, we see that, with the support of massively parallel environments, the RL method dREINFORCE has the same or better performance compared to the best-known solutions. PI-GNN does not achieve the best performance in any instance. We see that RL algorithms in Pattern II (e.g., dREINFORCE and MCPG) demonstrate much clearer advantages over the methods of Pattern I and conventional methods (e.g., Greedy, SA, and GA).

# C   TYPICAL CO PROBLEMS

- Partitioning problems: graph maxcut, graph partitioning, number partitioning. Graph maxcut and number partitioning are widely used in social networks (e.g., social influence maximization problem) which maximizes the social influence between two sets, and wireless mesh networks which maximize the total throughput between the transmitter set and the receiver set. Graph partitioning is often used in parallel computations, and the choice of partitioning strategies has strong impact on not only the performance of graph algorithms, but also the design of the algorithms.

- Covering problems: maximum independent set (MIS), knapsack, and set cover. MIS is widely used in chemical molecules analysis such as macromolecular docking: given two proteins, the protein docking problem is to find whether they interact to form a stable complex. Knapsack is widely used in stock trading, which maximizes the total value of the assets. Set cover is widely used in sensor networks which minimizes the number of sensors over all nodes.

- Coloring problems: graph coloring. It is used in routing and wavelength assignment problem, where light-path requests are given, generate a set of routing for each request and select a light-path per request and assign wavelengths to these selected paths.

- Mixed integer linear programming (MILP): binary integer linear programming (BILP). BILP is widely used in CO problem if all variables are binary, e.g., the assignment problem.

- Finance: portfolio optimization. It is widely used in finance, where we select a set of assets so that the total profits are maximized and the risk is minimized or constrained.

- Hamiltonian cycle: traveling salesman problem (TSP). It widely used in path planning problems such as designing a path for a taxi.

- Quantum: TNCO. It is a typical problem in quantum circuits, and it is also an important problem in quantum simulating by classical computers.

- Transportation: vehicle routing problem (VRP). It is widely used in transporting goods through multiple inventories.

## D BUILDING MASSIVELY PARALLEL ENVIRONMENTS ON GPUS

### D.1 PATTERN I FOR CO ENVIRONMENTS

In this subsection, we introduce three important functions of Pattern I: reset(), step(), and reward().

```python
def reset(self, graph_list):
    self.simulator = Simulator(graph_list=graph_list, device=self.device)
    xs = self.simulator.empty_xs()
    vs = reward(xs)
    return xs, vs
```

The "reset" function initializes a simulator, and sets the set as empty, i.e., $d(s_0)$ in Table 1. "graph_list" is a graph topology.

```python
def step(self, start_xs):
    xs = self.simulator.add(start_xs)
    vs = reward(xs)
    return xs, vs
```

In the "step" function, We add one node to the set in each environment, and the state changes to new ones, and the objective value is obtained. "start_xs" is the initial solutions over environments, and it returns the new solutions together with objective values.

```python
def reward(self, xs):
    num_sims = xs.shape[0]
    if num_sims != self.sim_ids.shape[0]:
        self.n0_ids = self.n0_ids[0].repeat(num_sims, 1)
        self.n1_ids = self.n1_ids[0].repeat(num_sims, 1)
        self.sim_ids = self.sim_ids[0:1] + th.arange(num_sims, dtype=self
                                          .int_type, device=self.
                                          device)[:, None]
    values = xs[self.sim_ids, self.n0_ids] ^ xs[self.sim_ids, self.n1_ids
                                          ]
    values = values.sum(1) // 2
    return values
```

In the "reward" function, we obtain the objective values of all environments given the states, i.e., $r$ in Table 1.

### D.2 PATTERN II FOR CO ENVIRONMENTS

In this subsection, we introduce three important functions of Pattern II: reset(), step(), and obj().

```
def reset(self, graph_list):
    self.simulator = Simulator(graph_list=graph_list, device=self.device)
    self.searcher = LocalSearch(simulator=self.simulator, num_nodes=self.
                                            num_nodes)
    good_xs = self.searcher.good_xs
    good_vs = self.searcher.good_vs
    return good_xs, good_vs
```

The "reset" function initializes random solutions, i.e., $d(s_0)$ in Table 1, and the RL algorithm searches better solutions based on them. "graph_list" is a graph topology, and the return is the good solutions with objective values. "Simulator" is a simulator for maxcut problem, and "LocalSearch" is a local search trick which returns the best neighborhood node within limited iterations.

```
def step(self, start_xs, probs):
    xs = metropolis_hastings_sampling(probs=probs, start_xs=start_xs,
                                        num_repeats=self.num_repeats,
                                        num_iters=-1)
    vs = self.searcher.reset(xs)
    for _ in range(self.num_searches):
        xs, vs, num_update = self.searcher.random_search(num_iters)
    return xs, vs
```

The "step" function returns new solutions based on current solutions, which includes two tricks: Metropolis Hastings (MH) sampling and local search. The MH sampling algorithm ensures the global balance in stochastic process, and make the MCMC simulations in stationary distributions. The local search helps the agent to search better solutions in neighborhoods."start_xs" is the initial solutions, and the "probs" are the probabilities obtained by the policy. "metropolis_hastings_sampling" is the MH sampling algorithm; "self.searcher" is the local search algorithm.

```
def obj(self, xs):
    num_sims = xs.shape[0]
    if num_sims != self.sim_ids.shape[0]:
        self.n0_ids = self.n0_ids[0].repeat(num_sims, 1)
        self.n1_ids = self.n1_ids[0].repeat(num_sims, 1)
        self.sim_ids = self.sim_ids[0:1] + th.arange(num_sims, dtype=self
                                            .int_type, device=self.
                                            device)[:, None]
    values = xs[self.sim_ids, self.n0_ids] ^ xs[self.sim_ids, self.n1_ids
                                            ]
    values = values.sum(1) // 2
    return values
```

The "obj" function returns the objective values of parallel environments given the states $xs$, i.e., $f$ in Table 1. All the solutions are stored by PyTorch tensors, thus the calculations are executed in parallel.

## E  LICENSE AND USAGE

The license is **MIT License**.

The following processes show how to run the algorithm.

1: select problem

config.py

```
PROBLEM = Problem.maxcut # We can select a problem such as maxcut.
```

2: select dataset Take methods/greedy.py as an example:

```
directory_data = '../data/syn_BA' # the directory of datasets
prefixes = ['barabasi_albert_100_'] # select the graphs with 100 nodes
```

3: run method

```
python methods/greedy.py    # run greedy

python methods/gurobiy.py    # run gurobi

python methods/simulated_annealing.py    # run simulated annealing

python methods/mcpg.py    # run mcpg

python methods/iSCO/main.py    # run iSCO

python methods/PI-GNN/main.py    # run PI-GNN

python methods/L2A/maxcut_end2end.py    # run ours
```

## F  ILP AND QUBO FORMULATIONS FOR 12 CO PROBLEMS

We show the basic denotations of graphs. Let $\mathcal{G} = (\mathcal{V}, \mathcal{E}, \boldsymbol{W})$ denote a weighted graph, where $\mathcal{V}$ is the node set, $\mathcal{E}$ is the edge set, $|\mathcal{V}| = V$, $|\mathcal{E}| = E$, and $\boldsymbol{W} : \mathcal{E} \to \mathbb{R}^+$ is the edge weight function, i.e., $\boldsymbol{W}_{u,v}$ is the weight of edge $(u, v) \in \mathcal{E}$. $\boldsymbol{W}_{u,v} > 0$ if $(u, v)$ is an edge and 0 otherwise. Let $\delta^+(i)$ and $\delta^-(i)$ denote the out-arcs and in-arcs of node $i$.

Integer linear programming (ILP) is a standard formulation of combinatorial optimization problems Ibaraki (1976). It has the *canonical form*:

$$\begin{aligned} \min \ &\boldsymbol{c}^T \boldsymbol{x} \\ \text{s.t. } &\boldsymbol{A}\boldsymbol{x} \leq \boldsymbol{b}, \\ &\boldsymbol{x} \geq 0, \\ &\boldsymbol{x} \in \mathbb{Z}^n, \end{aligned} \tag{6}$$

where $\boldsymbol{x}$ is a vector of $n$ decision variables, $\boldsymbol{c}$ is a vector of $n$ coefficients for $\boldsymbol{x}$ in the objective function, $\boldsymbol{A} \in \mathbb{R}^{m \times n}$ and $\boldsymbol{b} \in \mathbb{R}^m$ together denote $m$ linear constraints, and $\boldsymbol{x} \in \mathbb{Z}^n$ implies that we are interested in integer solutions. Let $\boldsymbol{x}^*$ denote the optimal solution and $f^*$ denote the corresponding objective value. Only a few problems such as portfolio optimization is quadratic programming, which will be described later.

With respect to QUBO or Ising model, we consider a 1D Ising model with a ring structure and an external magnetic field $\boldsymbol{h}_i$, there are $N$ nodes with $(N + 1) = 1 \mod N$; a node $i$ has a spin $\boldsymbol{s}_i \in \{+1, -1\}$ (where $+1$ for up and $-1$ for down). Two adjacent sites $i$ and $i + 1$ have an energy $\boldsymbol{w}_{i,i+1}$ or $-\boldsymbol{w}_{i,i+1}$ if they have the same direction or different directions, respectively.

The whole system will evolve into the ground state with the minimum Hamiltonian Cipra (1987) :

$$\arg\min_{\boldsymbol{s}} \ f(\boldsymbol{x}) = \underbrace{-\sum_{i=1}^{N} \boldsymbol{h}_i \boldsymbol{s}_i}_{f_A} + \alpha \underbrace{\sum_{i=1}^{N} -\boldsymbol{w}_{i,i+1} \boldsymbol{s}_i \boldsymbol{s}_{i+1}}_{f_B}, \tag{7}$$

where $\alpha$ is a weight, $f_A$ is defined on each node's effect on its own, and $f_B$ is defined on each two adjacent nodes' interactions. In fact, we generally use binary variables (0 or 1) to formulate the objective function, and $\boldsymbol{s}_i \in \{+1, -1\}$ can be replaced by

$$\boldsymbol{x}_i = \frac{\boldsymbol{s}_i + 1}{2}, \tag{8}$$

where $\boldsymbol{x}_i \in \{0, 1\}$.

### F.1  GRAPH MAXCUT

The graph maxcut problem is defined as follows. Given a graph $\mathcal{G} = (\mathcal{V}, \mathcal{E}, \boldsymbol{W})$, split $\mathcal{V}$ into two subsets $\mathcal{V}^+$ (with edge set $\mathcal{E}^+$) and $\mathcal{V}^-$ (with edge set $\mathcal{E}^-$), and the the cut set is $\delta = \{(i, j) | i \in \mathcal{V}^+, j \in \mathcal{V}^-\}$. The goal is to maximize the cut value: $\max \sum_{(i,j) \in \delta} \boldsymbol{W}_{i,j}$.

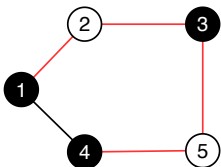

Figure 7: An example of graph maxcut.

### F.1.1   ILP FORMULATION

The ILP formulation of graph maxcut is:

$$\max \quad \sum_{(ij)} \boldsymbol{W}_{ij} \boldsymbol{y}_{ij} \tag{9}$$

$$\text{s.t.} \quad \boldsymbol{y}_{ij} \leq \boldsymbol{x}_i + \boldsymbol{x}_j, \forall i, j \in \mathcal{V}, i < j,$$
$$\boldsymbol{y}_{ij} \leq 2 - \boldsymbol{x}_i - \boldsymbol{x}_j, \forall i, j \in V, i < j,$$
$$\boldsymbol{y}_{ij} \geq \boldsymbol{x}_i - \boldsymbol{x}_j, \forall i, j \in V, i < j,$$
$$\boldsymbol{y}_{ij} \geq -\boldsymbol{x}_i + \boldsymbol{x}_j, \forall i, j \in V, i < j,$$

where $\boldsymbol{x}_i$ is a binary variable denoting if node $i$ belongs to the selected subset; and $\boldsymbol{y}_{ij}$ is 1 if nodes $i$ and $j$ are in different subsets and is 0 otherwise.

### F.1.2   QUBO FORMULATION

The QUBO formulation of graph maxcut is:

$$\min_{\boldsymbol{x}} f(\boldsymbol{x}) = -\frac{1}{2} \sum_{(i,j) \in \mathcal{E}} \boldsymbol{W}_{ij} \left(1 - (2\boldsymbol{x}_i - 1)(2\boldsymbol{x}_j - 1)\right), \tag{10}$$

where $\sum_{(i,j) \in \mathcal{E}} \boldsymbol{W}_{i,j}$ is a constant, $\boldsymbol{x}_i$ is 1 if node $i \in \mathcal{V}^+$, and 0 otherwise. The cut value $1 - (2\boldsymbol{x}_i - 1)(2\boldsymbol{x}_j - 1)$ is 1 if nodes $i$ and $j$ are in different subsets, and 0 otherwise.

For an illustrative example in the left graph of Fig. 7, the edge set is $E = \{(1,2), (1,4), (2,3), (2,4), (3,5)\}$ and the weights are $w_{1,2} = w_{1,4} = w_{2,3} = w_{2,4} = w_{3,5} = w_{4,5} = 1$. The edge set of black nodes is $\mathcal{E}^+ = \{(1,4)\}$, and the edge set of white nodes is $\mathcal{E}^- = \emptyset$. The edges connect the two subsets are $\delta = \{(1,2), (2,3), (2,4), (3,5), (4,5)\}$. The solution is $\boldsymbol{x} \in \{0,1\}^5$ and the Hamiltonian in (8) becomes

$$\begin{aligned}
\min_{\boldsymbol{x}} f(\boldsymbol{x}) \quad = \quad & -\left(\frac{1}{2} - \frac{1}{2}(2\boldsymbol{x}_1 - 1)(2\boldsymbol{x}_2 - 1)\right) - \left(\frac{1}{2} - \frac{1}{2}(2\boldsymbol{x}_1 - 1)(2\boldsymbol{x}_4 - 1)\right) \\
& -\left(\frac{1}{2} - \frac{1}{2}(2\boldsymbol{x}_2 - 1)(2\boldsymbol{x}_3 - 1)\right) - \left(\frac{1}{2} - \frac{1}{2}(2\boldsymbol{x}_2 - 1)(2\boldsymbol{x}_4 - 1)\right) \\
& -\left(\frac{1}{2} - \frac{1}{2}(2\boldsymbol{x}_3 - 1)(2\boldsymbol{x}_5 - 1)\right) - \left(\frac{1}{2} - \frac{1}{2}(2\boldsymbol{x}_4 - 1)(2\boldsymbol{x}_5 - 1)\right). \quad (11)
\end{aligned}$$

### F.1.3   EXAMPLE

**Pattern I** In left part of Fig. 8, the initial state is empty, i.e., no node is selected. Then we select node 1 with the maximum Q-value and add it to the state, thus the new state is [1]. The reward is 2.

**Pattern II** In right part of Fig. 8, the current state is [2, 3], i.e., node 2 and 3 are selected, and the objective value is 2. The new state is [1, 3, 4], i.e., node 1, 3, and 4 are selected, and the objective value is 4.

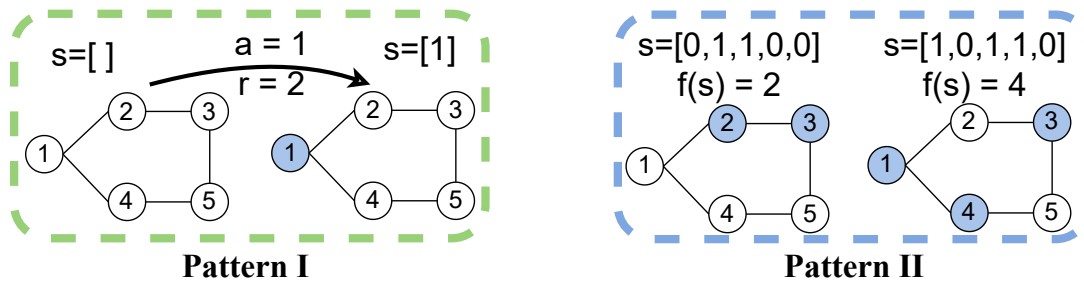

Figure 8: Two patterns for graph maxcut.

## F.2 GRAPH PARTITIONING

Given a graph, partition all nodes into two subsets ($\mathcal{V}^+$ and $\mathcal{V}^-$) of equal size $\frac{V}{2}$, such that the number of edges connecting the two subsets is minimized.

### F.2.1 ILP FORMULATION

The ILP formulation of graph partitioning is:

$$\min \quad \sum_{(i,j)\in\mathcal{E}} \boldsymbol{W}_{ij}\boldsymbol{y}_{ij} \tag{12}$$

$$\text{s.t.} \quad \boldsymbol{y}_{ij} \leq \boldsymbol{x}_i + \boldsymbol{x}_j, \forall i,j \in \mathcal{V}, i < j,$$
$$\boldsymbol{y}_{ij} \leq 2 - \boldsymbol{x}_i - \boldsymbol{x}_j, \forall i,j \in \mathcal{V}, i < j,$$
$$\boldsymbol{y}_{ij} \geq \boldsymbol{x}_i - \boldsymbol{x}_j, \forall i,j \in \mathcal{V}, i < j,$$
$$\boldsymbol{y}_{ij} \geq -\boldsymbol{x}_i + \boldsymbol{x}_j, \forall i,j \in \mathcal{V}, i < j,$$
$$\sum_{i\in\mathcal{V}} \boldsymbol{x}_i = \frac{V}{2},$$

where $\boldsymbol{x}_i$ is a binary variable denoting if node $i$ belongs to the selected subset; and $\boldsymbol{y}_{ij}$ is 1 if nodes $i$ and $j$ are in different subsets and is 0 otherwise. The first four constraints calculate the cut value $\boldsymbol{y}_{i,j}$ based on $\boldsymbol{x}_i$ and $\boldsymbol{x}_j$, and the last constraint makes sure that the nodes are partitioned into two subsets with equal size.

### F.2.2 QUBO FORMULATION

We consider a node $i \in \mathcal{V}$, and let $\boldsymbol{x}_i$ be a binary variable with +1 denoting in the subset $\mathcal{V}^+$ and 0 denoting in the subset $\mathcal{V}^-$. The Hamiltonian is

$$f_A = \left(\sum_{i\in\mathcal{V}} \boldsymbol{x}_i - \frac{V}{2}\right)^2, \tag{13}$$

$$f_B = \frac{1}{2} \sum_{(i,j)\in\mathcal{E}} \boldsymbol{W}_{ij} \left(1 - (2\boldsymbol{x}_i - 1)(2\boldsymbol{x}_j - 1)\right). \tag{14}$$

### F.2.3 EXAMPLE

**Pattern I** In left part of Fig. 9, the initial state is empty, i.e., no node is selected. Then we select node 2 and add it to the state, i.e., the new state is [2]. The reward is 1.

**Pattern II** In right part of Fig. 9, the current state is [2, 4], i.e., node 2 and 4 are put to the set, and the objective value is 4. The new state is [1, 2], i.e., node 1 and 2 are put to the set, and the new objective value is 2.

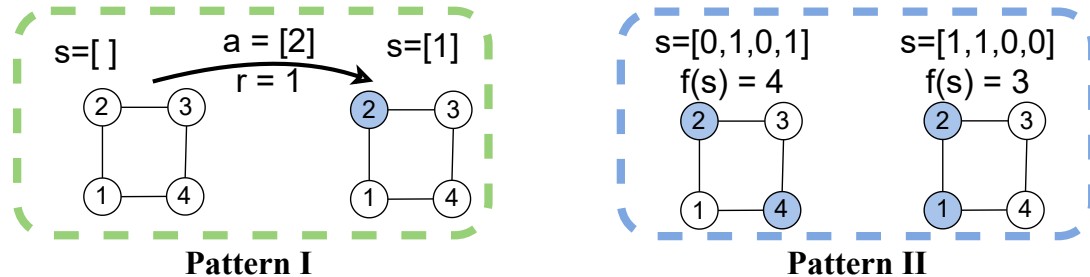

Figure 9: Two patterns for graph partitioning.

## F.3   NUMBER PARTITIONING

Given a set of $I$ positive numbers $\mathcal{I} = \{n_1, \ldots, n_I\}$, partition this set of numbers into two disjoint subsets $\mathcal{S}$ and $\mathcal{I}\backslash\mathcal{S}$, such that the discrepancy between the sum of elements in both sets is minimized.

### F.3.1   ILP FORMULATION

The ILP formulation of number partitioning is

$$\min \quad y \tag{15}$$
$$\text{s.t.} \quad \sum_{i \in \mathcal{I}} n_i \boldsymbol{x}_i - \sum_{i \in \mathcal{I}} n_i(1 - \boldsymbol{x}_i) \geq y,$$
$$\sum_{i \in \mathcal{I}} n_i \boldsymbol{x}_i - \sum_{i \in \mathcal{I}} n_i(1 - \boldsymbol{x}_i) \leq -y,$$
$$\boldsymbol{x}_i \in \{0, 1\} \,\forall i \in \mathcal{I}, y \geq 0, \tag{16}$$

where $\boldsymbol{x}_i$ is a binary variable with 1 denoting the $i$-th element is in the selected subset $\mathcal{S}$, and 0 otherwise, and $y$ is the absolute value of the discrepancy between the sum of elements in both sets.

### F.3.2   QUBO FORMULATION

The QUBO formulation of number partitioning is

$$\min \quad f = \left( \sum_{i \in \mathcal{I}} n_i(2\boldsymbol{x}_i - 1) \right)^2 \tag{17}$$

where $\boldsymbol{x}_i$ is a binary variable with 1 denoting the $i$-th element is in the selected subset $\mathcal{S}$, and 0 otherwise.

### F.3.3   EXAMPLE

Let [1, 3, 5, 8, 9] be a list of numbers.

**Pattern I** The initial state is empty, i.e., no number is selected. Then we select number 5 and add it to the state, i.e., the new state is [5]. The reward is 16.

**Pattern II** The current state is [0, 0, 1, 0, 1], i.e., number 5 and 9 are in a set, and others are put into another set. The new state is [0, 0, 1, 1, 0], i.e., remove number 9 and add number 8 to the set. The reward is 4.

## F.4   BILP

The binary integer linear programming (BILP) aims to obtain the max value an objective function with equation constraints, where all variables are binary. We assume there are $I$ binary variables with indices $\mathcal{I} = \{1, \ldots, I\}$ and $J$ constraints with indices $\mathcal{J} = \{1, \ldots, J\}$.

### F.4.1 ILP FORMULATION

The ILP formulation of BILP is

$$
\max \quad \sum_{i \in \mathcal{I}} \boldsymbol{c}_i \boldsymbol{x}_i \tag{18}
$$

$$
\text{s.t.} \quad \sum_{i \in \mathcal{I}} \boldsymbol{s}_{j,i} \boldsymbol{x}_i = \boldsymbol{b}_j, \forall j \in \mathcal{J},
$$

where $\boldsymbol{x}_i$ is a binary variable (1 or 0), $\boldsymbol{c}_i$ is a coefficient, and $\boldsymbol{b}_j$ is a right-hand constant.

### F.4.2 QUBO FORMULATION

The QUBO formulation of BILP is

$$
\min f = -\sum_{i \in \mathcal{I}} \boldsymbol{c}_i \boldsymbol{x}_i + \alpha \sum_{j \in \mathcal{J}} \left( \boldsymbol{b}_j - \sum_{i \in \mathcal{I}} s_{j,i} \boldsymbol{x}_i \right)^2 , \tag{19}
$$

where $\alpha \gg 1$ is a positive constant.

### F.4.3 EXAMPLE

Take a BILP as an example:

$$
\begin{aligned}
\max \ & 3\boldsymbol{x}_1 + 2\boldsymbol{x}_2 \\
\text{s.t.} \ & 2\boldsymbol{x}_1 + 3\boldsymbol{x}_2 \leq 4, \\
& \boldsymbol{x}_1, \boldsymbol{x}_2 \in \{0, 1\}.
\end{aligned} \tag{20}
$$

**Pattern I** The initial state is empty, i.e., $\boldsymbol{x}_1 = \boldsymbol{x}_2 = 0$. Then we select $\boldsymbol{x}_1$ and add it to the state, i.e., the new state is [1]. The reward is 3.

**Pattern II** The current state is [0, 0], i.e., $\boldsymbol{x}_1 = \boldsymbol{x}_2 = 0$, and the objective value is 0. The new state is [0, 1], i.e., $\boldsymbol{x}_1 = 0, \boldsymbol{x}_2 = 1$, and the objective value is 2.

## F.5 PORTFOLIO OPTIMIZATION

We consider $n$ assets $\boldsymbol{S}_1, \ldots, \boldsymbol{S}_n$ with future returns $\boldsymbol{r} = [\boldsymbol{r}_1, \ldots, \boldsymbol{r}_n]$. The portfolio is denoted by $\boldsymbol{x} = [\boldsymbol{x}_1, \ldots, \boldsymbol{x}_n]$. To ensure the number of assets to hold is $k$, we introduce a binary vector $\boldsymbol{y} = [\boldsymbol{y}_1, \ldots, \boldsymbol{y}_n]$. The symmetric covariance matrix of the returns of assets is denoted by $\boldsymbol{A}$. The aim is to minimize the variance with ensuring the total return.

### F.5.1 QUADRATIC PROGRAMMING FORMULATION

The quadratic programming (QP) formulation of portfolio optimization is

$$
\begin{aligned}
\min \quad & \boldsymbol{x}^T \boldsymbol{A} \boldsymbol{x} \\
\text{s.t.} \quad & \boldsymbol{r}^T \boldsymbol{x} \geq R_{\min}, \\
& \mathbf{1}^T \boldsymbol{x} = 1, \\
& \boldsymbol{x} \leq B \boldsymbol{y}, \\
& \mathbf{1}^T \boldsymbol{y} = k, \\
& \boldsymbol{x} \geq 0, \boldsymbol{y} \in \{0, 1\},
\end{aligned} \tag{21}
$$

where the superscript $T$ denotes the transpose of a vector or matrix, $R_{\min}$ is the minimum return of all assets, $\mathbf{1}$ is a column of with all elements being 1, and $B$ is a large positive number.

### F.5.2 QUBO FORMULATION

The QUBO formulation of portfolio optimization is

$$\min \quad \boldsymbol{x}^T \boldsymbol{A} \boldsymbol{x} - \alpha \, \text{sgn}(\boldsymbol{r}^T \boldsymbol{x} - R_{\min}) + \alpha \, (\boldsymbol{1}^T \boldsymbol{x} - 1)^2 + \alpha \, \text{sgn}(\boldsymbol{x} - \boldsymbol{B}\boldsymbol{y})^2 + \alpha \, (\boldsymbol{1}^T \boldsymbol{y} - k)^2, \tag{22}$$

where $\text{sgn}(z)$ is a sign function with 1 if $z >= 0$ and 0 otherwise, and $\alpha$ is a large positive constant to ensure the constraints are satisfied.

### F.5.3 EXAMPLE

Take the following as an example:

$$\begin{aligned}
\min \quad & 2\boldsymbol{x}_1^2 + \boldsymbol{x}_2^2 \\
\text{s.t.} \quad & \boldsymbol{x}_1 + \boldsymbol{x}_2 \geq 0, \\
& \boldsymbol{x}_1 + \boldsymbol{x}_2 = 1, \\
& \boldsymbol{x}_1 \leq B\boldsymbol{y}_1, \\
& \boldsymbol{x}_2 \leq B\boldsymbol{y}_2, \\
& \boldsymbol{y}_1 + \boldsymbol{y}_2 = 1, \\
& \boldsymbol{x}_1, \boldsymbol{x}_2 \geq 0, \boldsymbol{y}_1, \boldsymbol{y}_2 \in \{0, 1\},
\end{aligned} \tag{23}$$

where $B$ is a large positive value.

**Pattern I** The initial state is empty, i.e., $\boldsymbol{x}_1 = \boldsymbol{x}_2 = 0$, and then we select $\boldsymbol{x}_1$ and add it to the state, i.e., the new state is [1]. The reward is $B - 2$ since the constraint is satisfied now.

**Pattern II** The current state is [0, 1], and the objective value is 1. The new state is [1, 0], and the objective value is 2.

### F.6 MIS

An independent set $\mathcal{S} \subset \mathcal{V}$ is a set of mutually non-adjacent nodes of the graph. The maximum independent set (MIS) problem aims to obtain the largest independent set.

### F.6.1 ILP FORMULATION

The ILP formulation of MIS is

$$\begin{aligned}
\max \quad & \sum_{i \in V} \boldsymbol{x}_i \\
\text{s.t.} \quad & \boldsymbol{x}_i + \boldsymbol{x}_j \leq 1, \forall (i, j) \in \mathcal{E},
\end{aligned} \tag{24}$$

where $\boldsymbol{x}_i$ is a binary variable with 1 denoting in the independent set, and 0 otherwise.

### F.6.2 QUBO FORMULATION

If we write the QUBO formulation of MIS as

$$\min H = -\sum_{i \in V} \boldsymbol{x}_i - \alpha \sum_{(i,j) \in \mathcal{E}} (2 - \boldsymbol{x}_i - \boldsymbol{x}_j)^2, \tag{25}$$

where $\alpha \gg 1$ is a constant. However, it does not work since the penalties are different when $\boldsymbol{x}_i + \boldsymbol{x}_j = 0$ and $\boldsymbol{x}_i + \boldsymbol{x}_j = 1$. To make the penalties equal under the two cases, we have to introduce the item $\gamma \alpha (1 - \boldsymbol{x}_i - \boldsymbol{x}_j)^2$ where $\gamma$ is unknown and should be calculated. We list the values of all cases of $(\boldsymbol{x}_i, \boldsymbol{x}_j)$: Let $\alpha(\gamma - 3) = 0$, and we obtain $\gamma = 3$. Therefore, the correct QUBO formulation of MIS is

$$\min -\sum_{i \in \mathcal{V}} \boldsymbol{x}_i - \alpha \sum_{(i,j) \in \mathcal{E}} (2 - \boldsymbol{x}_i - \boldsymbol{x}_j)^2 + 3\alpha \sum_{(i,j) \in \mathcal{E}} (1 - \boldsymbol{x}_i - \boldsymbol{x}_j)^2, \tag{26}$$

Table 8: All cases of $(\boldsymbol{x}_i, \boldsymbol{x}_j)$ in MIS.

| $\boldsymbol{x}_i$ | $\boldsymbol{x}_j$ | $-\alpha(2 - \boldsymbol{x}_i - \boldsymbol{x}_j)^2$ | $\gamma\alpha(1 - \boldsymbol{x}_i - \boldsymbol{x}_j)^2$ |
|---|---|---|---|
| 0 | 0 | $-4\alpha$ | $\gamma\alpha$ |
| 0 | 1 | $-\alpha$ | 0 |
| 1 | 0 | $-\alpha$ | 0 |
| 1 | 1 | 0 | $\gamma\alpha$ |

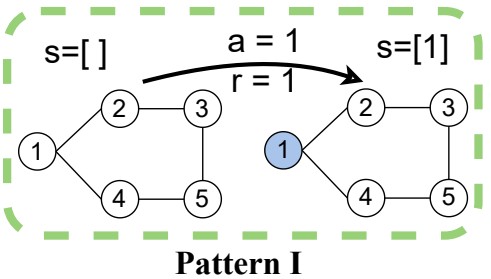
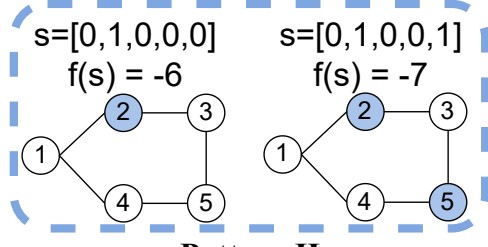

**Pattern I**                                 **Pattern II**

Figure 10: Two patterns for MIS.

### F.6.3 EXAMPLE

**Pattern I** In left part of Fig. 10, the initial state is empty, i.e., no node is selected. Then we select node 1 with the maximum Q-value and add it to the state, i.e., the new state is [1]. The reward is 1.

**Pattern II** In right part of Fig. 10, the current state is [0, 1, 0, 0, 0], which only includes node 2, and the objective value is -6. The new state is [0, 1, 0, 0, 1], which includes node 2 and 5, and the objective value is -7.

### F.7 KNAPSACK

**Knapsack problem**. Given a set of items $\mathcal{I}$, each item $i$ with an integer weight $\boldsymbol{W}_i$ and a value $\mu_i$, determine which items to include in the collection so that the total weight is less than or equal to a given limit $U$ and the total value is maximized.

### F.7.1 ILP FORMULATION

The ILP formulation of Knapsack is

$$
\begin{aligned}
\max \quad & \sum_{i \in \mathcal{I}} \mu_i \boldsymbol{x}_i \\
\text{s.t.} \quad & \sum_{i \in \mathcal{I}} \boldsymbol{W}_i \boldsymbol{x}_i \leq U,
\end{aligned}
\tag{27}
$$

where $\boldsymbol{x}_i$ is a binary variable (1 denoting in the knapsack, and 0 otherwise), and $U$ is a constant.

### F.7.2 QUBO FORMULATION

We consider an item $i$, and let $\boldsymbol{x}_i$ be a binary variable with 1 denoting in the knapsack and 0 otherwise. Let $\boldsymbol{y}_n$ for $1 \leq n \leq U$ be a binary variable with 1 denoting the final weight of the knapsack is $n$ and 0 otherwise. The QUBO formulation of Knapsack is

$$
\min_x f = \left( \sum_{n=1}^{U} \boldsymbol{y}_n \right)^2 + \left( \sum_{n=1}^{U} n\boldsymbol{y}_n - \sum_{i \in \mathcal{I}} \boldsymbol{W}_i \boldsymbol{x}_i \right)^2 - \alpha \sum_{i \in \mathcal{I}} \mu_i \boldsymbol{x}_i,
\tag{28}
$$

where $\alpha > 0$ is a weight. The first two items should not be violated; therefore, we require that $0 < \alpha \max(\mu_i) < 1$.

### F.7.3 EXAMPLE

We assume there are 3 items, and their values are [1, 1, 2], and their weights are [1, 1, 1], and the limit of total weight is 2.

**Pattern I** The initial state is empty. Then we select item 2 and add it to the state, i.e., the new state is [2]. The reward is 1.

**Pattern II** The current state is [0, 1, 0], and the objective value is 1. The new state is [0, 1, 1], and the objective value is 3.

## F.8 SET COVER

**Set cover problem**: Given a finite set $\mathcal{U}$ and subsets $\mathcal{V}_i \subseteq \mathcal{U}$ ($i \in \mathcal{I} = \{1, \ldots, I\}$), find the smallest number of subsets so that the union of them is $\mathcal{U}$, i.e., $\bigcup_{i \in \mathcal{I}} \mathcal{V}_i = \mathcal{U}$.

### F.8.1 ILP FORMULATION

The ILP formulation of set cover problem is

$$\min \quad \sum_{i \in \mathcal{I}} \boldsymbol{x}_i \tag{29}$$
$$\text{s.t.} \quad \sum_{i:u \in \mathcal{V}_i} \boldsymbol{x}_i \geq 1, \forall u \in \mathcal{U},$$

where $\boldsymbol{x}_i$ is a binary variable with 1 denoting the subset $\mathcal{S}_i$ is selected, and 0 otherwise.

### F.8.2 QUBO FORMULATION

Let $\boldsymbol{y}_{u,m}$ be a binary variable with 1 denoting the number of subsets which include element $u$ is $m \geq 1$, and 0 otherwise. Let $\boldsymbol{x}_i$ be a binary variable with 1 denoting the subset $\mathcal{V}_i$ is selected, and 0 otherwise. The QUBO formulation of set cover is

$$\min_{x} H = \sum_{i \in \mathcal{I}} \boldsymbol{x}_i + \alpha \sum_{u \in \mathcal{U}} \left(1 - \sum_{m \in \mathcal{I}} \boldsymbol{y}_{u,m}\right)^2 + \alpha \sum_{u \in \mathcal{U}} \left(\sum_{m \in \mathcal{I}} m \boldsymbol{y}_{u,m} - \sum_{i:u \in \mathcal{V}_i} \boldsymbol{x}_i\right)^2, \tag{30}$$

where $\alpha > 1$ is a constant. The first item minimizes the total number of selected subsets; the second item means that exactly one $\boldsymbol{y}_{u,m}$ over $m$ must be 1 since each element of $\mathcal{U}$ must be selected a fixed number of times; the third item ensures that the number of times $u$ is included is equal to the number of $\mathcal{V}_i$ which is included.

### F.8.3 EXAMPLE

We assume the universe set $U = \{1, 2, 3, 4, 5\}$ and the collection of sets $S = \{\{1, 2, 3\}, \{2, 4\}, \{4, 5\}\}$.

**Pattern I** The current state is [4, 5]. Then we select the first set $\{1, 2, 3\}$ and add it to the state, i.e., the new state is [1, 2, 3, 4, 5]. The reward is $B - 2$ since they cover all elements of the universe set.

**Pattern II** We assume $\alpha = 10$ in equation 30. The current state is [0, 1, 0], the objective value is 11 since it cannot cover all the elements. The new state is [1, 0, 1], and the objective value is 2 since they cover all the elements.

## F.9 GRAPH COLORING

Given a graph and $I$ colors where each color $i$ with a weight $\boldsymbol{W}_i$, color each vertex with a specific color using the minimum sum of weights, such that no edge connects two vertices with the same color.

### F.9.1 ILP FORMULATION

The ILP formulation of graph coloring problem is

$$\min \quad \sum_{i \in \mathcal{I}} \boldsymbol{W}_i \sum_{v \in \mathcal{V}} \boldsymbol{x}_{vi} \tag{31}$$

$$\text{s.t.} \quad \sum_{i \in \mathcal{I}} \boldsymbol{x}_{vi} = 1, \forall v \in \mathcal{V},$$

$$\boldsymbol{x}_{ui} + \boldsymbol{x}_{vi} \leq 1, \forall (u, v) \in \mathcal{E},$$

where $\boldsymbol{x}_{vi}$ is a binary variable with 1 denoting vertex $v$ is colored using the $i$-th color, and 0 otherwise.

### F.9.2 QUBO FORMULATION

The QUBO formulation of graph coloring problem is

$$\min_x f = \sum_{i \in \mathcal{I}} \boldsymbol{W}_i \sum_{v \in \mathcal{V}} \boldsymbol{x}_{vi} + B \sum_{v \in \mathcal{V}} \left( 1 - \sum_{i \in \mathcal{I}} \boldsymbol{x}_{vi} \right)^2 + B \sum_{(u,v) \in \mathcal{E}} \sum_{i \in \mathcal{I}} \boldsymbol{x}_{ui} \boldsymbol{x}_{vi}, \tag{32}$$

where $B \gg \boldsymbol{W}_i$ is a positive constant, and $\boldsymbol{x}_{v,i}$ is a binary variable with 1 denoting vertex $v$ is colored using the $i$-th color, and 0 otherwise.

### F.9.3 EXAMPLE

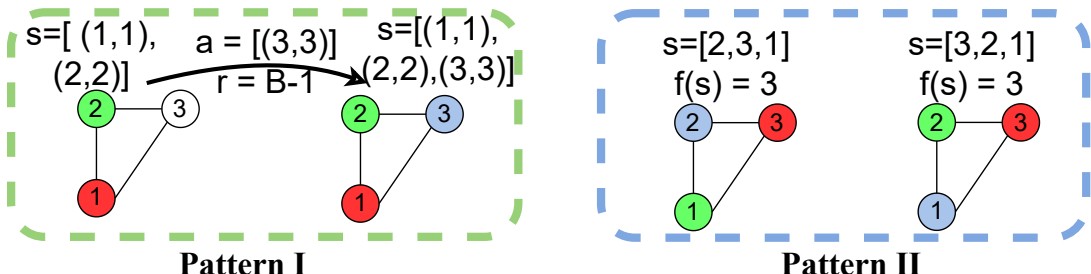

Figure 11: Two patterns for graph coloring.

We assume color 1, 2, and 3 are denoted by red, green, and blue, respectively.

**Pattern I** In left part of Fig. 11, the current state is [(1,1),(2,2)], i.e., node 1 is colored by color 1 (red), and node 2 is colored by color 2 (green). Then we color node 3 by color 3 (blue) and add it to the state. The new state is [(1,1),(2,2), (3,3)]. The number of used colors increases by 1, thus the reward is $B - 1$.

**Pattern II** In right part of Fig. 11, the current state is [2, 3, 1], the objective value is 3. The new state is [3, 2, 1], and the objective value is still 3 since the number of used colors does not changed.

### F.10 TSP

**Traveling salesman problem (TSP)**. Find a tour with the minimum total weights, where the tour is a Hamilton cycle that visits each node of the graph exactly once.

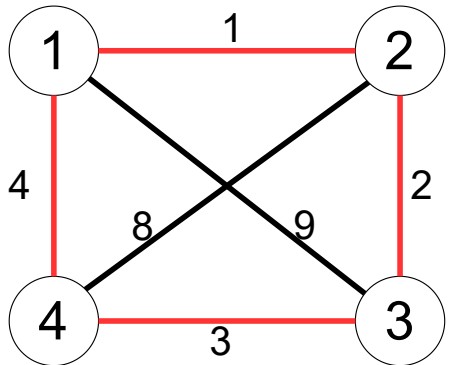

Figure 12: An example of TSP

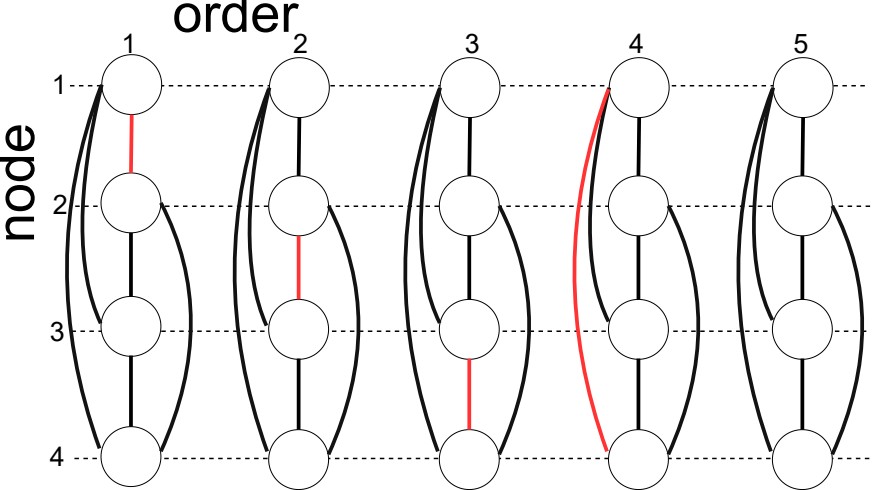

Figure 13: QUBO of TSP.

### F.10.1 ILP FORMULATION

The ILP formulation of TSP problem is

$$
\begin{aligned}
\min \quad & \sum_{(u,v)\in\mathcal{E}} \boldsymbol{W}_{uv}\boldsymbol{x}_{uv}, && (33)\\
\text{s.t.} \quad & \sum_{v\in\mathcal{V},(u,v)\in\mathcal{E}} \boldsymbol{x}_{u,v} = 1, u \in \mathcal{V},\\
& \sum_{u\in\mathcal{V},(u,v)\in\mathcal{E}} \boldsymbol{x}_{uv} = 1, v \in \mathcal{V},\\
& \boldsymbol{t}_u + \boldsymbol{W}_{uv} \le \boldsymbol{t}_v + B(1 - \boldsymbol{x}_{uv}), \forall(u,v) \in \mathcal{E},\\
& \boldsymbol{x} \in \{0,1\}, \boldsymbol{t} \ge 0,
\end{aligned}
$$

where $\boldsymbol{x}_{uv} = 1$ if edge $(u,v)$ is passed and 0 otherwise, and $\boldsymbol{t}_u$ denotes the arrival time at node $u$, and $B$ is a big positive value. The first and second set of constraints ensure each node is visited exactly once, and the third set of constraints is the subtour elimination.

### F.10.2 QUBO FORMULATION

Let $\boldsymbol{x}_{ij}$ be 1 if node $i$ appears with the order $j$ in the cycle and 0 otherwise. The Hamiltonian is

$$\min_{\boldsymbol{x}} f_A = \sum_{i=1}^{n}\left(1 - \sum_{j=1}^{n} \boldsymbol{x}_{ij}\right)^2 + \sum_{j=1}^{n}\left(1 - \sum_{i=1}^{n} \boldsymbol{x}_{ij}\right)^2, \tag{34}$$

$$\min_{\boldsymbol{x}} f_B = \sum_{(u,v)\notin\mathcal{E}} \sum_{j=1}^{n} \boldsymbol{x}_{uj}\boldsymbol{x}_{vj+1} + \alpha \sum_{(u,v)\in\mathcal{E}} \boldsymbol{W}_{uv} \sum_{j=1}^{n} \boldsymbol{x}_{uj}\boldsymbol{x}_{vj+1}, \tag{35}$$

where $\alpha > 0$. $f_A$ ensures that each node appears once in the cycle. For $f_B$, the first item provides a penalty if $(i, j)$ is not the edge, and the second item is the total weights of edges in the cycle.

As shown in Fig. 12, the edge set is $E = \{(1, 2), (1, 3), (1, 4), (2, 3), (2, 4), (3, 4)\}$, and the weights $\boldsymbol{W}_{1,2} = \boldsymbol{W}_{2,1} = 1, \boldsymbol{W}_{1,3} = \boldsymbol{W}_{3,1} = 9, \boldsymbol{W}_{1,4} = \boldsymbol{W}_{4,1} = 4, \boldsymbol{W}_{2,3} = \boldsymbol{W}_{3,2} = 2, \boldsymbol{W}_{2,4} = \boldsymbol{W}_{4,2} = 8, \boldsymbol{W}_{3,4} = \boldsymbol{W}_{4,3} = 3$. The solution at the $k$-th iteration is $\boldsymbol{x}^k \in \{-1, +1\}^5$ and its Hamiltonian becomes

$$\min_{\boldsymbol{x}^k} f(\boldsymbol{x}^k) = \alpha(\boldsymbol{x}_{1,1}^k \boldsymbol{x}_{2,2}^k + 2\boldsymbol{x}_{2,2}^k \boldsymbol{x}_{3,3}^k + 3\boldsymbol{x}_{3,3}^k \boldsymbol{x}_{4,4}^k + 4\boldsymbol{x}_{4,4}^k \boldsymbol{x}_{1,5}^k). \tag{36}$$

### F.10.3 EXAMPLE

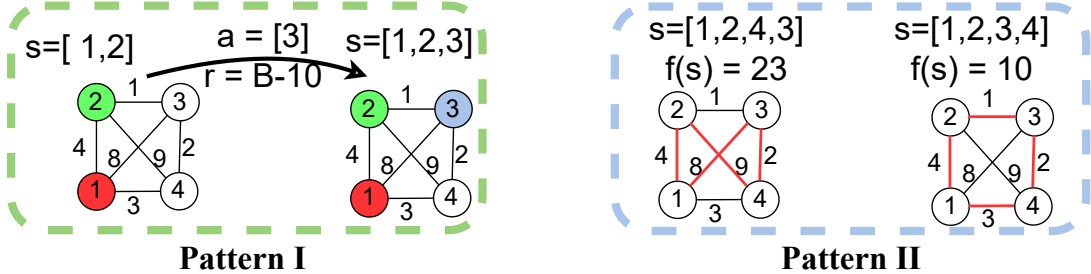

Figure 14: Two patterns for TSP.

**Pattern I** In left part of Fig. 14, the current state is $[1, 2]$, i.e., edge $(1, 2)$ is passed. Then we select node 3 and add it to the state, i.e., the new state is $[1, 2, 3]$. Node 4 is clearly the last node. These nodes constitute a Hamiltonian cycle, thus the reward is $B - 10$. The objective value, i.e., the total distance, is 10.

**Pattern II** In right part of Fig. 14, the current state is $[1,2,4,3]$, i.e., the path is $1 \Longrightarrow 2 \Longrightarrow 4 \Longrightarrow 3 \Longrightarrow 1$, and the objective value is 23. The new state is $[1,2,3,4]$, i.e., the path is $1 \Longrightarrow 2 \Longrightarrow 3 \Longrightarrow 4 \Longrightarrow 1$, and the objective value is 10.

### F.11 TNCO

Given a graph $\mathcal{G}$ representing a tensor network, where a node represents a tensor, an edge represents a contraction operation, and the weight is the contraction cost defined as the number of multiplications. Starting with $\mathcal{G}_1 = \mathcal{G}$, a contraction ordering path $\boldsymbol{P} = (\boldsymbol{e}_1, \ldots, \boldsymbol{e}_{N-1})$ will generate a sequence of graphs $(\mathcal{G}_1, \ldots \mathcal{G}_{N-1}, \mathcal{G}_N)$, where $\mathcal{G}_t$ with $\mathcal{E}_t$ is the $t$-th tensor network, $\mathcal{G}_N$ has only one tensor, and $\boldsymbol{e}_t \in \mathcal{E}_t, t = 1, \ldots, N - 1$. The goal is to find a contraction ordering path $P$ with minimum cost.

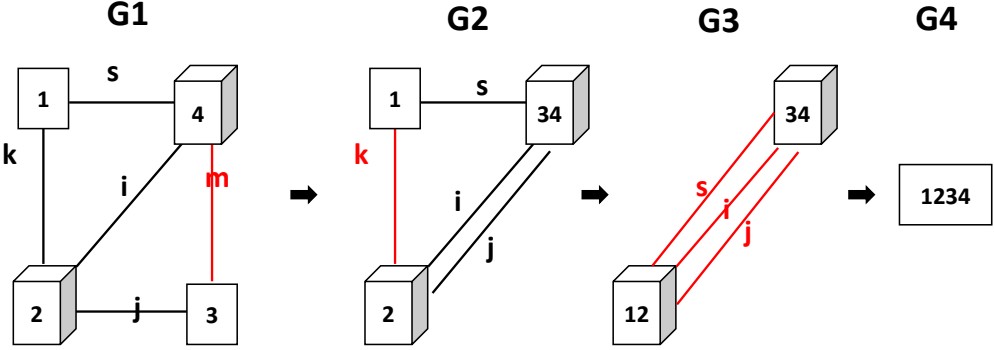

Figure 15: An example of TNCO.

### F.11.1 ILP FORMULATION

The ILP formulation of TNCO is

$$\boldsymbol{P}^*(G) = \arg\min_P \sum_{t=1}^{N-1} \boldsymbol{W}_t(\boldsymbol{e}_t), \tag{37}$$
$$\text{s.t. } \boldsymbol{P} = (\boldsymbol{e}_1, \ldots, \boldsymbol{e}_{N-1}), t = 1, ..., N-1,$$

where $\boldsymbol{W}_t(\boldsymbol{e}_t)$ is defined as the number of multiplications for the tensor contraction along edge $\boldsymbol{e}_t$. Fig. 15 shows a TNCO example with corresponding contacted graphs, where a red solid line denotes a contraction of two nodes. Graph $G_1$ has $V_1 = (1, 2, 3, 4), E_1 = (k, j, m, i, s), \boldsymbol{W} = \{K, J, M, I, S\}$. Assuming that the first contraction operation is on index $m$ in $G_1$, tensors 3 and 4 are contracted into tensor 34 at a computation cost of $IJMS$ multiplications. Then, the graph is updated to $G_2$ with $V_2 = (1, 2, 34), E_2 = (k, ij, s), \boldsymbol{W} = \{K, IJ, S\}$. Assuming the second contraction operation is on index $k$ in $G_2$, tensor 1 and 2 are contracted into tensor 12 at a computation cost of $SKIJ$ multiplications. The updated graph $G_3$ has $V_3 = (12, 34), E_3 = (sij), \boldsymbol{W} = \{sij\}$. Finally, tensors 12 and 34 are contracted into a real number using $SIJ$ multiplications. The total number of multiplications is $IJMS + SKIJ + SIJ$.

### F.11.2 QUBO FORMULATION

The QUBO formulation of TNCO problem uses $N(N-1)$ spins $\boldsymbol{x}_{u,j}$, where $u$ denotes the tensor and $j$ denotes its order in the TNCO path. We use $\boldsymbol{J}_{u,v}^i$ to denote the cost introduced by the tensor contraction between $u$ and $v$ for the $i$-th order, *i.e.*, the number of multiplications. The energy of the original TNCO problem has three terms. The first term requires there are exactly two tensors selected at order $j$ along a path. The second term measures the contraction cost at order $j$. These are encoded in the following Hamiltonian:

$$f(\boldsymbol{x}) = \sum_{i=1}^{N-1} \left\{ (2 - \sum_{u=1}^{N-i} \boldsymbol{x}_{u,i})^2 + \sum_{u=1}^{N} \sum_{v=1}^{N} \boldsymbol{J}_{u,v}^i \boldsymbol{x}_{u,i} \boldsymbol{x}_{v,i} \right\}. \tag{38}$$

As shown in the right graph in Fig. 16, a red solid line denotes the contraction of two nodes, and the Hamiltonian is $f(x) = (2 - \boldsymbol{x}_{1,1} - \boldsymbol{x}_{2,1} - \boldsymbol{x}_{3,1} - \boldsymbol{x}_{4,1})^2 + (2 - \boldsymbol{x}_{1,2} - \boldsymbol{x}_{2,2} - \boldsymbol{x}_{3,2})^2 + (2 - \boldsymbol{x}_{1,3} - \boldsymbol{x}_{2,3})^2 + \boldsymbol{W}_1(1,4)\boldsymbol{x}_{1,1}\boldsymbol{x}_{4,1} + \boldsymbol{W}_1(1,2)\boldsymbol{x}_{1,1}\boldsymbol{x}_{2,1} + \boldsymbol{W}_1(2,3)\boldsymbol{x}_{2,1}\boldsymbol{x}_{3,1} + \boldsymbol{W}_1(2,4)\boldsymbol{x}_{2,1}\boldsymbol{x}_{4,1} + \boldsymbol{W}_1(3,4)\boldsymbol{x}_{3,1}\boldsymbol{x}_{4,1} + \boldsymbol{W}_2(1,3)\boldsymbol{x}_{1,2}\boldsymbol{x}_{3,2} + \boldsymbol{W}_2(1,2)\boldsymbol{x}_{1,2}\boldsymbol{x}_{2,2} + \boldsymbol{W}_2(2,3)\boldsymbol{x}_{2,2}\boldsymbol{x}_{3,2} + \boldsymbol{W}_3(1,2)\boldsymbol{x}_{1,3}\boldsymbol{x}_{2,3}.$

### F.11.3 EXAMPLE

**Pattern I** In left part of Fig. 17, the initial state is empty. Then we select the edge (3, 4) and add it to the state, thus the new state is [(3, 4)]. The reward is 48.

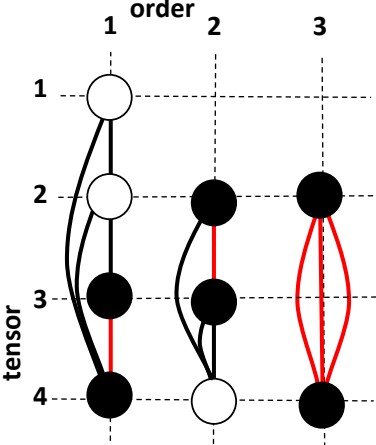

Figure 16: QUBO formulation of TNCO.

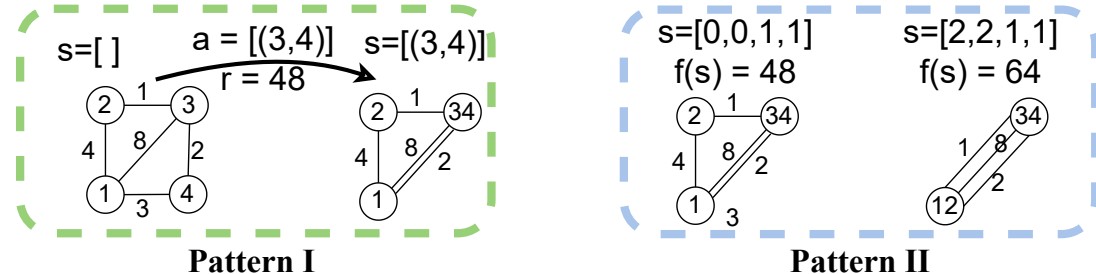

Figure 17: Two patterns for TNCO.

**Pattern II** In right right of Fig. 17, node 3 and node 4 are with the order 1, thus the current state is [0, 0, 1, 1] and the objective value is 48. Then node 1 and node 2 are with the order 2, thus the new state is [2, 2, 1, 1], and the objective value is 64.

## F.12  VRP

There some goods in a depot denoted by point 0, given a set of $N$ other points $\mathcal{N} = \{1, 2, \ldots, N\}$, referred as customers. Each customer $i$ has a demand $q_i \geq 0$. There are a set of vehicles $\mathcal{K} = \{1, 2, \ldots, K\}$ with the same capacity $Q > 0$. We define $q_0 = 0$. Design each route for each vehicle so that each vehicle starts from the depot and finally return to the depot after visiting customers, and each customer is visited by exactly one vehicle.

### F.12.1 ILP FORMULATION

We introduce binary variables $\boldsymbol{x}_{i,j}$ $((i,j) \in \mathcal{E})$ indicating how often a vehicle directly moves from $i$ to $j$. The ILP formulation of VRP is

$$\min \quad \sum_{(i,j)\in\mathcal{E}} \boldsymbol{W}_{ij}\boldsymbol{x}_{ij}, \tag{39}$$

$$\text{s.t.} \quad \sum_{j\in\delta^+(i)} \boldsymbol{x}_{ij} = 1, i \in \mathcal{N},$$

$$\sum_{i\in\delta^-(j)} \boldsymbol{x}_{ij} = 1, j \in \mathcal{N},$$

$$\sum_{j\in\delta^+(0)} \boldsymbol{x}_{0j} = K,$$

$$\sum_{(i,j)\in\delta^+(S)} \boldsymbol{x}_{ij} \geq r(\mathcal{S}), \mathcal{S} \subseteq \mathcal{N}, \mathcal{S} \neq \emptyset,$$

$$\boldsymbol{x}_{ij} \in \{0,1\}, \forall (i,j) \in \mathcal{E},$$

where $r(\mathcal{S})$ is the minimum number of vehicle routes needed to serve $\mathcal{S}$. The first constraint indicates that each customer has one successor. The second constraint indicates that each customer has one predecessor. The third constraint means that $K$ routes are constructed. The fourth constraint is serve at the capacity constraints and the subtour elimination constraints.

### F.12.2 QUBO FORMULATION

The QUBO formulation of VRP is

$$\min_x f = \sum_{(i,j)\in\mathcal{E}} \boldsymbol{W}_{ij}\boldsymbol{x}_{ij} + B\sum_{i\in\mathcal{N}}\left(\sum_{j\in\delta^+(i)} \boldsymbol{x}_{ij} - 1\right)^2 + B\sum_{j\in\mathcal{N}}\left(\sum_{i\in\delta^-(j)} \boldsymbol{x}_{ij} - 1\right)^2$$

$$+ B\left(\sum_{j\in\delta^+(0)} \boldsymbol{x}_{0j} - K\right)^2 - B\sum_{\mathcal{S}\subseteq\mathcal{N},\mathcal{S}\neq\emptyset} \text{sgn}\left(\sum_{(i,j)\in\delta^+(S)} \boldsymbol{x}_{ij} - r(\mathcal{S})\right), \tag{40}$$

where $\text{sgn}(z)$ is a sign function with 1 if $z >= 0$ and 0 otherwise, and $B$ is a large positive constant to ensure the constraints are satisfied.

### F.12.3 EXAMPLE

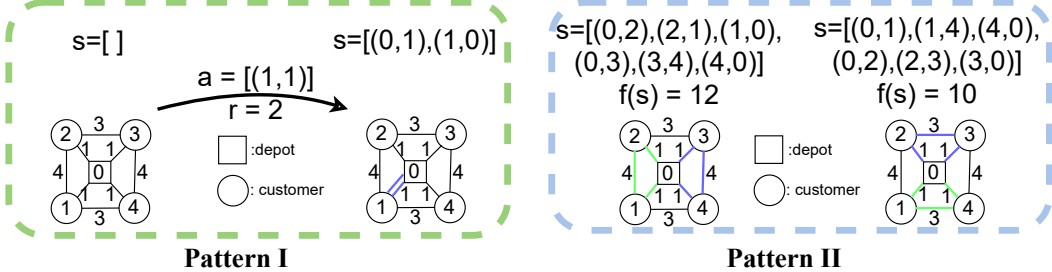

Figure 18: Two patterns for VRP.

In Fig. 18, the depot is denoted by a square, and a customer is denoted by a circle.

**Pattern I** In left part of Fig. 18, the initial state is empty. Then we select node 1 for vehicle 1, i.e., the action is (1, 1), and add it to the state, and the new state is [(0,1),(1,0)], and the reward is 2.

**Pattern II** We assume the path for vehicle 1 is denoted by the green line, and the path for vehicle 2 is denoted by the blue line. In right part of Fig. 18, the current state is [(0,2),(2,1),(1,0), (0,3),(3,4),(4,0)] since the path for vehicle 1 is $0 \Longrightarrow 2 \Longrightarrow 1 \Longrightarrow 0$, and the path for vehicle 2 is $0 \Longrightarrow 3 \Longrightarrow 4 \Longrightarrow 0$, and the objective value is 12. The new state is [(0,1),(1,4),(4,0),(0,2),(2,3),(3,0),] since the path for vehicle 1 is $0 \Longrightarrow 1 \Longrightarrow 4 \Longrightarrow 0$ and the path for vehicle 2 is $0 \Longrightarrow 2 \Longrightarrow 3 \Longrightarrow 0$, and the objective value is 10.