# OpenReview forum: "Massively Parallel Environments for Large-Scale Combinatorial Optimizations Using Reinforcement Learning"
_ICLR.cc/2025/Conference — ICLR 2025 Conference Withdrawn Submission_

### Official Review · Reviewer_Mqub · 2024-10-24

**Soundness:** 2
**Presentation:** 2
**Contribution:** 2
**Rating:** 3
**Confidence:** 4

**Summary:**

The paper introduces a novel approach to massively parallel environments for large-scale combinatorial optimization (CO) problems using reinforcement learning (RL). The authors identify sampling speed as a significant bottleneck in applying RL to CO problems and propose a solution using GPU-based massively parallel environments. They argue that this approach offers several advantages over traditional CPU-based methods, including increased parallelism, reduced communication overhead between CPUs and GPUs, and the ability to train RL-based methods on large-scale CO problems.

The effectiveness of their approach is demonstrated by constructing 24 GPU-based massively parallel environments for 12 CO problems, which are used to train RL-based methods. They reproduce benchmark RL algorithms, including both instance-wise and distribution-wise approaches, and test them on synthetic and real-world datasets. The results show significant improvements in both sampling speed and training efficiency, with the capability to train on problems involving thousands of nodes.

**Strengths:**

The paper's strength lies in its use of GPU-based massively parallel environments to accelerate the sampling process for large-scale combinatorial optimization (CO) problems using reinforcement learning (RL) algorithms. By leveraging GPUs, the authors achieve a significant speedup in sampling compared to conventional CPU-based methods, enabling the training of RL agents on larger CO problems than was previously possible.

**Weaknesses:**

Although the paper addresses 12 combinatorial optimization (CO) problems, it essentially only focuses on the MaxCut problem. While the appendix provides formulations of other CO problems in ILP and QUBO, the comparisons with other solvers are primarily conducted for the MaxCut problem. Therefore, the contribution of this paper to the CO field is limited.

The authors claim that most CO problems can be formulated in QUBO under Pattern II, which allows for a wider range of applications. However, when CO problems with complex and numerous constraints are formulated in QUBO, parameters must be assigned to each constraint and incorporated into the objective function. Setting these parameters appropriately is challenging, often making it difficult to even find a feasible solution, let alone the optimal one. Therefore, except for simple CO problems like MaxCut, which have no constraints, it cannot be said that QUBO formulations are effective for general CO problems.

For MaxCut and QUBO, the following are high-performance solvers that generate good solutions quickly, even for practical-sized problems. Thus, while it may be difficult to directly compare their performance, some mention of the following should be included in the paper:
Daniel Rehfeldt, Thorsten Koch, and Yuji Shinano, Faster Exact Solution of Sparse MaxCut and QUBO Problem, Mathematical Programming Computation (MPC), Volume 15, pages 445–470, (2023).

There is little description of the implementation on GPUs and the computational performance of the GPUs. This raises questions about whether the use of 24 expensive NVIDIA A100 GPUs is essential, or if similar performance could be achieved with fewer GPUs through optimizations in GPU implementation. Moreover, when comparing with Gurobi, it is difficult to make a fair evaluation since the computing environments differ significantly (Gurobi does not use GPUs).

In summary, my opinions are as follows:

1: While creating a high-performance solver for the MaxCut problem is meaningful, the performance for other CO problems is unclear, limiting its contribution to the optimization field.

2: The MaxCut problem is a very simple CO without any constraints. Other CO problems, however, are more complex and have numerous constraints, making it difficult to even find a feasible solution when converting them into QUBO form. Therefore, there is a substantial gap between being able to formulate a problem as QUBO and actually solving it as QUBO.

3: If the MaxCut problem is to be addressed, it should be compared not only with Gurobi but also with other methods (exact and approximate solutions).

4: It is necessary to discuss whether 24 GPUs are genuinely required or if similar results could be achieved with fewer computational resources. Since Gurobi uses only CPUs, the computational resources differ in this study’s comparison experiments, and it cannot simply be concluded that the proposed method is superior.

**Questions:**

Please provide any counterarguments or additional points regarding the weaknesses mentioned above.

---

### Official Review · Reviewer_2fGJ · 2024-11-03

**Soundness:** 3
**Presentation:** 3
**Contribution:** 3
**Rating:** 6
**Confidence:** 3

**Summary:**

This paper build 24 GPU-based massively parallel environments for 12 CO problems. Specifically, the authors reproduce benchmark RL algorithms, including instance-wise and distribution-wise approaches especially in large-scale CO problems, on both synthetic datasets and real-world datasets. Experiments demonstrate that the benchmark significantly improves the sampling speed, the scale of trained problems, and the objective value.

**Strengths:**

1.	Developing a comprehensive combinatorial optimization benchmark for RL-based approaches is important for the community. Specifically, this paper first build 24 GPU-based massively parallel environments for 12 CO problems, and then reproduce benchmark RL algorithms, including instance-wise and distribution-wise approaches on both synthetic datasets and real-world datasets.
2.	Experiments demonstrate that the benchmark significantly improves the sampling speed, the scale of trained problems, and the objective value. Specifically, the proposed benchmark improves the sampling speed by two orders, the objective value obtained under certain cases is on par with the state-of-the-art solver Gurobi, which is the golden solver across the world.

**Weaknesses:**

1.	The technical contribution of the benchmark seems limited. Specifically, this paper build 24 GPU-based massively parallel environments for 12 CO problems, and reproduce benchmark existing RL algorithms on both synthetic datasets and real-world datasets. It would be more valuable if the authors could propose some new RL algorithms based on their extensive benchmark results.
2.	Although the benchmark enhances the scale of trained problems by an order of magnitude to thousands of nodes, it still remains relatively small compared to real-world industrial problems with tens of thousands of nodes.
3.	The authors claim that they evaluate the RL algorithms on real-world datasets. However, I do not find the description of real-world datasets in Section 5. It would be more convincing if the authors could clarify what real-world datasets they use.

**Questions:**

Please refer to weaknesses for details.

---

### Official Review · Reviewer_wFdk · 2024-11-04

**Soundness:** 2
**Presentation:** 2
**Contribution:** 2
**Rating:** 3
**Confidence:** 2

**Summary:**

The paper explores the use of Reinforcement Learning (RL) for Combinatorial Optimization (CO) and identifies sampling speed as a major bottleneck in applying RL to CO problems. To address this, the authors propose leveraging GPUs to improve sampling efficiency. The paper contributes to the field by parallelizing both instance-wise and distribution-wise approaches and implementing two RL-based algorithms for CO. The authors create 24 GPU-based "massively parallel" environments to tackle 12 CO problems, providing benchmarks on both real and synthetic datasets. The results show a significant increase in sampling speed, improving solution discovery rates by orders of magnitude. Empirically, the solutions obtained are accurate in terms of objective value and closely match those produced by state-of-the-art (SOTA) commercial solvers.

**Strengths:**

S1. The paper identifies sampling speed as a bottleneck in applying reinforcement learning (RL) to combinatorial optimization (CO) and proposes massively parallel environments with two distinct "patterns" for implementing RL-based algorithms.

S2. According to the authors, this is the first work to introduce massively parallel RL environments for the distribution-wise approach, which broadens the applicability of the methodology to a wider range of CO problems.

S3. The paper experimentally demonstrates the flexibility of the Pattern I and II approaches by implementing 11 different algorithms, providing an extensive comparison of their performance across various methods.

**Weaknesses:**

W1. The proposal of massively parallel environments for reinforcement learning (RL) is not novel, as similar approaches have been explored in prior work [1, 2, 3, 4]. Additionally, the "patterns" discussed for combinatorial optimization (CO) environments have previously been covered in the literature. This raises questions about the exact technical contributions of the paper, which require a clearer and more detailed explanation. The authors claim to extend prior methods by incorporating a distribution-wise approach, yet there is insufficient experimental evidence to demonstrate the approach's effectiveness across different CO problems.

W2. The novelty of the work is unclear. The paper does not adequately explain how these GPU-based massively parallel environments were constructed or how the parallelization of different patterns was achieved. Furthermore, the motivation behind implementing Pattern II, the challenges encountered, and the reasons prior work could not address these challenges are not well highlighted. The results obtained seem to be an artifact of using GPU, but details (if they exist) about the technical advancements made to deploy the algorithms on the GPU have a lack of detail.

W3. The paper describes two patterns, I and II, with Pattern II reportedly providing higher solution quality at the cost of slower processing due to more complex sampling methods (What are they, I have also asked that in the question section). This trade-off may limit the speed advantages of the approach when high-quality solutions are prioritized. It would be beneficial to present objective values versus time for both patterns, preferably within a single plot for direct comparison.

Citations:

[1] Nair, A., Srinivasan, P., Blackwell, S., Alcicek, C., Fearon, R., De Maria, A., ... & Silver, D. (2015). Massively parallel methods for deep reinforcement learning. arXiv preprint arXiv:1507.04296.

[2] Clemente, A. V., Castejón, H. N., & Chandra, A. (2017). Efficient parallel methods for deep reinforcement learning. arXiv preprint arXiv:1705.04862.

[3] Khalil, E., Dai, H., Zhang, Y., Dilkina, B., & Song, L. (2017). Learning combinatorial optimization algorithms over graphs. Advances in neural information processing systems, 30.

[4] Berto, F., Hua, C., Park, J., Luttmann, L., Ma, Y., Bu, F., ... & Park, J. (2023). Rl4co: an extensive reinforcement learning for combinatorial optimization benchmark. arXiv preprint arXiv:2306.17100.

**Questions:**

Major Questions:

Q1. “Moreover, we use several tricks to improve the quality of solutions, e.g., sampling algorithms.”.
What tricks are employed to improve the quality of solutions?

Q2. “Moreover, from experiments, we see that the methods in Pattern II are generally better than that in Pattern I.” Do you anticipate it to work on almost all the CO problems? Do you have any theoretical insights/results to back this claim?

Q3. Where can I find results for the other 12 CO problems? I only see results for max-cut with different datasets.

Q4. What algorithm does Gurobi specifically use to solve CO problems used in your benchmarks?

Q5. Is “dREINFORCE” the method that was contributed in this case? I saw it was linked with external citations, so I am not sure if it was borrowed as a massively parallel environment framework to implement Pattern II. Please clarify.

Q6. Can you summarize the novel discoveries and insights that this paper presents?

Minor Issues:

1. The paper does not have line numbers in the left margin.

2. The header contains “Under Review as a conference paper at ICLR 2024”, which should be “...ICLR 2025”.

3. I suggest using \citet for in-text citations (using the citation in a sentence) and \citep for parenthetical citations, i.e., where you just want the citation as a reference.

---

### Official Review · Reviewer_fge5 · 2024-11-04

**Soundness:** 2
**Presentation:** 2
**Contribution:** 2
**Rating:** 5
**Confidence:** 3

**Summary:**

This paper proposes a new RL environment for large-scale combinatorial optimization problems, which implementes GPU-accelerated parrallel sampling, and implements both instance-wise and distribution-wise approaches.

**Strengths:**

1. This work builds 24 enviornments for 12 different CO problems, including many commonly-used benchmarking problems.
1. The implemented enviornment can effectively improve the training speed of RL approaches by accelerating the GPU-based sampling.

**Weaknesses:**

1. Discussions on the previous RL enviornment [Ecole](https://doc.ecole.ai/py/en/stable/index.html), which is a usually used enviornment for MILPs, is missing.
1. This work mainly focuses on CO problems on graphs, rather than general MILP or QUBO problems, which limits the application scope.
1. The considered two patterns mainly involve approaches those predicting scores for each node on a graph. However, some other approaches, such as cut sselections, are not considered.
1. It is not introduced how can a user define a new enviornment with a new CO problem not included. I think it is importent that a user can easily transfer this enviornment to a new problem.
1. The citations are all in a wrong form, "Author (Year)" rather than "(Author, Year)".
1. I think this is a valuable work from the perspective of engineering implementation. However, its technical novelty as a research paper is still not very significant. The diffuculty in implementing such a parallel system is not demonstrated enough.

**Questions:**

See weaknesses.

---

### Comment · Area_Chair_bsP1 · 2024-11-21
**No author response yet**

Dear Submission381 Authors,

ICLR encourages authors and reviewers to engage in asynchronous discussion up to the 26th Nov deadline. It would be good if you can post your responses to the reviews soon.

---

### Note · Authors · 2024-11-24

I have read and agree with the venue's withdrawal policy on behalf of myself and my co-authors.